# Towards Efficient Post-Training Quantization For Large Vision-Language Models Via Token-Wise Redundancy Elimination

## Abstract

Post-training quantization (PTQ) has emerged as an effective technique for compressing large models and accelerating inference without retraining. While PTQ has been extensively studied in large language models (LLMs), its application to vision-language models (VLMs) remains underexplored. In this work, we identify two intrinsic characteristics of VLM activations: 1) visual over-representation, where vision tokens are excessive and often redundant, and 2) modality gap, which refers to the clear separation between text and vision tokens in the latent feature space. Together, these two factors significantly deteriorate quantization performance but have been overlooked by existing PTQ methods. To address these challenges, we propose VLMQ, A VLM-tailored PTQ framework that selectively prioritizes salient tokens while suppressing redundant ones during quantization. In particular, we introduce a gradient-driven importance factor to capture the token-wise importance variance, the effectiveness of which is substantiated through both empirical and theoretical analysis. To ensure efficiency, we propose to use lightweight block-wise backpropagation for factor acquisition. Finally, we reformulate the optimization objective into an importance-aware form to preserve importance activation information. Extensive evaluations on 8 benchmarks across 0.5B∼32B VLMs demonstrate the state-of-the-art (SOTA) performance of our VLMQ, particularly under low-bit settings. For example, it achieves a substantial **16.45%** improvement on MME-RealWorld under 2-bit quantization. Code is provided in the supplementary material.

## 1 Introduction

Large language models (LLMs) (Bubeck et al., 2023; Touvron et al., 2023a;b; DeepSeek-AI, 2025; Jiang et al., 2023) have demonstrated exceptional advancements across diverse natural language processing tasks, leading to an increased emphasis on developing vision-language models (VLMs) (Wang et al., 2024b; Bai et al., 2025; Li et al., 2024a; Chen et al., 2024d; Xiaomi, 2025; Zhu et al., 2025) that process multi-modal inputs, including texts, images, and videos. Despite their impressive capabilities, the unprecedented scaling in the model size complicates their deployment in practical resource-limited contexts.

In light of these problems, quantization has provided an effective solution by converting full-precision weights and activations (*e.g.*, FP16/BF16) into reduced-precision formats (*e.g.*, INT8/INT4), thereby markedly lowering memory footprint and computation complexity. To be more specific, post-training quantization (PTQ) has emerged as a prevalent approach for deploying large-scale models, owing to its minimal computational overhead and the ability to bypass the costly fine-tuning or retraining process. Substantial research efforts have been directed toward designing advanced PTQ methods tailored to LLMs. Such studies typically aim to refine the distributions of weights and activations through various strategies, including equivalent transformations (Lin et al., 2023; Xiao et al., 2023), Hessian-based error compensation (Frantar & Alistarh, 2022; Li et al., 2025b), and incoherence processing (Ashkboos et al., 2024; Hu et al., 2025). While significant progress has been achieved in applying PTQ to LLMs, their extension to VLMs has not yet been adequately investigated. Recent pioneering research efforts, such as MBQ (Li et al., 2025a), MQuant (Yu et al., 2025), and QSLAW (Xie et al., 2024), emphasize the modality imbalance challenge and

propose dedicated importance-aware strategies to enhance the performance of quantized VLMs. However, these methods either require expensive parameter fine-tuning (Xie et al., 2024), specialized manipulation at inference time (Yu et al., 2025), or rely on a suboptimal grid search (Li et al., 2025a), failing to offer an efficient and effective solution for both calibration and inference.

In this work, we highlight two intrinsic properties of VLMs that critically affect quantization performance. In particular, VLM activations reveal (i) a pronounced visual over-representation (i.e., limited text tokens vs. excessive and redundant vision tokens) as well as (ii) a modality gap. However, existing PTQ approaches for LLMs minimize the layer-wise reconstruction loss while treating all tokens uniformly (Lin et al., 2023; Frantar et al., 2022; Li et al., 2025b), without accounting for token-level informativeness or importance. Such a token-agnostic design inevitably biases the quantized

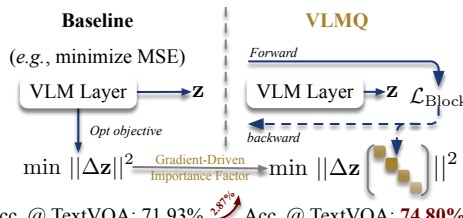

**Figure 1:** Baseline vs. VLMQ. The diagonal matrix in VLMQ represents the importance factors. The reported accuracy is from 2-bit Qwen2-VL-7B-Instruct (Wang et al., 2024b) quantized by GPTQ and VLMQ.

model toward dominant but redundant visual features, thereby yielding significant performance drops. Consequently, directly transferring these methods to VLMs is suboptimal.

To alleviate the adverse effects of visual over-representation and modality gap, we propose VLMQ, an importance-aware calibration framework specifically designed for VLMs. We introduce an importance factor $\mathbf{G}$ to capture token-wise variations in informativeness. To obtain effective importance factors, we first formulate a theoretical and empirical connection between loss perturbation and token-level quantization error. Guided by this Theorem, we further adopt a lightweight block-wise backpropagation strategy that efficiently derives gradient-driven, layer-specific importance factors. Finally, as shown in Figure 1, VLMQ refines the conventional optimization objective into an importance-aware formulation, assigning greater weight to salient tokens while down-weighting redundant ones.

The core contributions are summarized as follows:

- We uncover a fundamental mismatch between the vision redundancy inherent in VLMs and the token-agnostic objectives of existing mainstream PTQ approaches, which treat all tokens uniformly during quantization. We verify that this neglected redundancy disrupts layer reconstruction, thereby accounting for the degraded performance observed when directly applying LLM PTQ methods to VLMs.
- We propose a gradient-driven importance factor $\mathbf{G}$ to suppress less informative tokens. Its effectiveness is theoretically and experimentally supported through the established link between loss perturbation and first-order token-level activation errors, while its efficient computation is ensured via a lightweight block-wise backpropagation scheme.
- Through extensive experiments, we validate the superior performance of our framework on VLMs, particularly under ultra-low-bit quantization regimes. Notably, our approach achieves state-of-the-art (SOTA) results and yields accuracy improvements of up to 16.45%.

## 2 BACKGROUND

### 2.1 PRELIMINARY

**Notation.** $N$, $C_i$, $C_o$, and $\mathcal{V}$ denote the sequence length, in-channel number (hidden size), out-channel number (intermediate size), and the vocabulary set. We adopt the bold lowercase and uppercase letters to represent row vectors and matrices, respectively. The letters with a hat symbol (e.g., $\hat{\mathbf{x}}$ or $\hat{\mathbf{W}}$) represent the quantized weights or activations noised by quantization errors. The linear operation is described as $\mathbf{Y} = \mathbf{W}\mathbf{X}$ where $\mathbf{W} \in \mathbb{R}^{C_o \times C_i}$ and $\mathbf{X} \in \mathbb{R}^{C_i \times N}$ represent the weight and activation. The indexing rule is that $\mathbf{W}_{j,:}$ and $\mathbf{W}_{:,j}$ indicate the $j$-th row and $i$-th column of matrix $\mathbf{W}$ respectively. A negative index signifies the removal of a row in a matrix (e.g., $\mathbf{X}_{-1} \in \mathbb{R}^{(C_i-1) \times N}$).

**VLM architecture.** The advanced development of VLMs (Wang et al., 2024b; Bai et al., 2025; Chen et al., 2024d; Li et al., 2024a) handles data in different modalities, including text, image, and video. The VLMs comprise three key components: a visual encoder, a vision-text projector, and a language model (LM) backbone. The mainstream decoder-only LM backbones output a distribution $\mathbf{Prob} = (p_1, p_2, ..., p_{N-1}) \in \mathbb{R}^{N \times |\mathcal{V}|}$ and refer to it to generate predictions via diverse decoding strategies (Stern et al., 2018; Xia et al., 2022; Kim et al., 2022).

We define input for each decoding layer in the above-mentioned LM backbone as activation $\mathbf{X} \in \mathbb{R}^{C_i \times N}$. The activation transforms across $L$ decoding layers in LM backbones. The attention stream and feed-forward network stream within one decoding layer are defined as

$$\texttt{Attn}(\mathbf{X}) = \mathbf{X} + \texttt{MHSA}(\mathbf{X}), \tag{1}$$

$$\texttt{MLP}(\mathbf{X}) = \mathbf{X} + \texttt{FFN}(\mathbf{X}), \tag{2}$$

where $\texttt{MHSA}(\cdot)$ describes the multi-head self-attention operation with Q/K/V/O projections and $\texttt{FFN}(\cdot)$ represents the feed-forward data flow with Up/Gate/Down projections. We omit the layer normalization involved in decoding layers for simplicity.

## 2.2 RELATED WORK

**Advanced VLMs.** Building upon the rapid advancements in LLMs, VLMs have demonstrated exceptional capabilities in visual understanding (Li et al., 2024a; Wang et al., 2024b; Bai et al., 2025; Xiaomi, 2025; Chen et al., 2024d; Zhu et al., 2025). For instance, Qwen2-VL (Wang et al., 2024b) enhances multimodal representations through techniques such as M-RoPE and naive dynamic resolution, facilitating unified text-image-video comprehension with flexible visual tokenization. Qwen2.5-VL (Bai et al., 2025) further refines VLM capabilities by enabling omnidocument parsing, supporting ultra-long video understanding, and enhancing agent functionalities across both desktop and mobile platforms, thereby exhibiting advanced vision-language intelligence. InternVL3 (Zhu et al., 2025) contributes to the evolution of multimodal large language models (MLLMs) by introducing variable visual position encoding for handling extended contexts. Leveraging these innovations, it excels in multimodal perception, reasoning, and textual performance.

**PTQ for large models.** PTQ has become a widely adopted technique for compressing (Li & Panda, 2024; Wnag et al., 2024; Huang et al., 2025a; Gong et al., 2024) and accelerating large models (Lv et al., 2024; Chen et al., 2024a; Tian et al., 2024; Huang et al., 2025b). In the context of LLMs, approaches such as AWQ (Lin et al., 2023) and SmoothQuant (Xiao et al., 2023) address activation outliers through smooth-based transformations. Hessian-informed methods (Frantar & Alistarh, 2022; Frantar et al., 2022; Li et al., 2025b) provide closed-form, layer-wise quantization schemes. More recently, rotation-based methods (Ashkboos et al., 2024) have improved quantization robustness by reshaping parameter distributions, thereby improving low-bit quantization performance. However, the PTQ becomes challenging for VLMs due to dissimilar features and distributions between text and vision modality, and remains underexplored. MQuant (Yu et al., 2025) proposes modality-specific static quantization and attention-invariant flexible switching to address the modality gap and visual outliers. MBQ (Li et al., 2025a) observes error-sensitivity variance across different modalities. Built on Lin et al. (2023), an improved scale searching strategy is proposed that takes modality variance into consideration. QSLAW (Xie et al., 2024) determines the quantization step size via a learning-based method and proposes modality-aware warmup to alleviate the overfitting issue. Unlike these methods concentrating on modality difference, Q-VLM (Wang et al., 2024a) captures cross-layer dependency in VLMs and partitions blocks based on entropy.

## 3 MOTIVATION

VLMs process inputs from both text and vision modalities, which exhibit significantly different statistical properties and distributional characteristics. We identify two key observations (*i.e.*, visual over-representation and modality gap) that together have a substantial impact on the performance of quantized models. In this section, we first introduce these two observations and discuss the insights they provide. Then, we present a pilot study to empirically validate these findings.

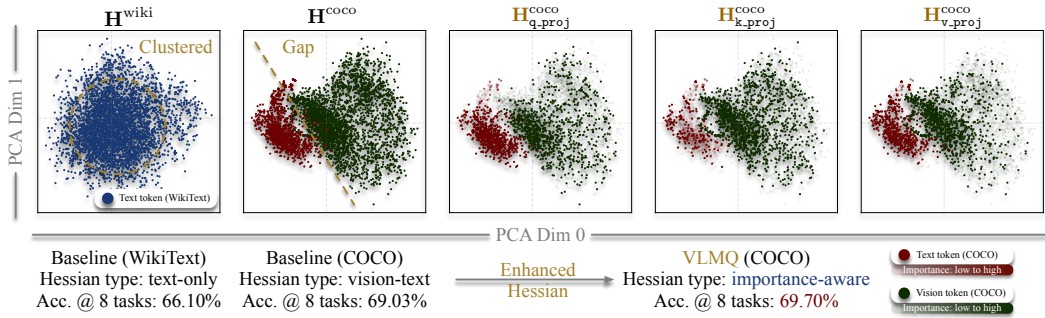

**Figure 2:** PCA-based (Shlens, 2014) activation feature analysis with activations (4096 points) extracted from the pre-attention breakpoint of the 20-th transformer layer in Qwen2-VL-7B-Instruct. The left two subfigures depict the activation feature distributions constructed from text-only and mixed text-vision activations, respectively. The right three subfigures visualize the distributions with varying token-wise importance factors. Light red/green and dark red/green points denote tokens classified as important or unimportant ones. Reported average accuracy is under INT3 quantization across eight vision-language benchmarks.

### 3.1 Modality Discrepancy

**Observation 1: visual over-representation.** We have found that VLM inputs contain an excessive number of *redundant* vision tokens, which dominate the activation space (see Figure 2). We term this phenomenon as *visual over-representation*. Current PTQ approaches originally designed for LLMs overlook this redundancy. They minimize the layer-wise mean square error (MSE) by treating all tokens equally, following a standard optimization formulation (Lin et al., 2023; Frantar et al., 2022; Li et al., 2025b; Shao et al., 2023).

**Observation 2: modality gap.** We also uncover a modality gap in VLMs, which denotes the distinct separation between **text tokens** and **vision tokens** in the latent feature space, as shown in Figure 2. Such misalignment can bias the calibration process, especially for existing studies (Lin et al., 2023; Frantar et al., 2022), which disproportionately favor excessive and redundant vision tokens over informative ones during quantization. As a result, existing token-agnostic methods often incur non-negligible accuracy drops in VLMs (see Section 5).

### 3.2 Pilot Study

**Insight: down-weighting vision tokens improves quantization.** These observations motivate the need for importance-aware quantization strategies that explicitly distinguish salient vision tokens from redundant ones. We hypothesize that *assigning lower weights to a subset of vision tokens balances the distribution density across modalities*, thereby enhancing quantization performance. To validate this hypothesis, we conduct the following pilot study.

**Pilot study on vision-role in quantization.** To examine the impact of vision tokens on quantization, we conduct a pilot study assessing how they affect model performance. We benchmark INT3-quantized VLMs on DocVQA (Mathew et al., 2021) by randomly down-weighting a subset of vision tokens with a low importance factor. As shown in Table 1, increasing the LI ratio within a certain range alleviates performance degradation. Notably, performance reaches its peak when 50% of vision tokens are marked as low-importance (LI), suggesting that maintaining a balanced level of visual input is crucial for effective quantization. To better explain this

**Table 1:** Statistics under controlled settings where a random subset of vision tokens are manually assigned as low-importance (LI) tokens. ♣ denotes LI tokens are down-weighted by a factor 0.01.

| *Qwen2-VL-7B-Instruct-INT3* | | |
|---|---|---|
| **Calib Modality** | **LI Ratio♣** | **DocVQA Acc** |
| Text-only | - | 86.86% |
| Text-vision | 0% | 88.09% |
| Text-vision | 25% | 88.19% |
| Text-vision | 50% | 88.48% |
| Text-vision | 75% | 87.87% |
| Text-vision (Ours) | Fine-grained | **88.90%** |

phenomenon, in Figure 2, we further color-code token points by their importance factors $\mathbf{G}$ (defined later) in the right three subfigures. The evident imbalance in importance density across modalities risks skewing the distribution toward redundant visual margins, rather than concentrating on the more informative central regions. These findings highlight the necessity of fine-grained importance-aware quantization to counteract redundancy-induced bias.

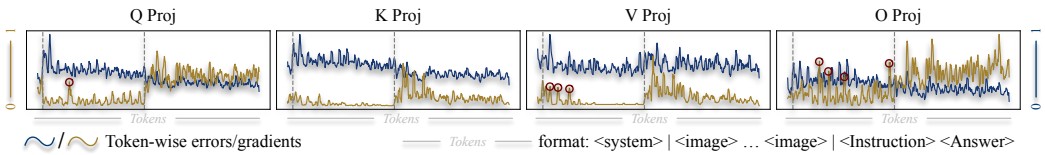

**Figure 3:** Visualization of normalized token-wise error ($\Delta \mathbf{z}$) and gradient ($\mathbf{p}^{(\Delta \mathbf{z})}$). The red circle indicates the salient vision tokens. The magnitude of the error remains relatively stable across tokens, whereas the gradient varies across tokens in different modalities.

Overall, the observed performance peak under moderate down-weighting of vision tokens underscores the importance of maintaining a balanced level of inputs. These findings further motivate the development of a *fine-grained* and *importance-aware* quantization strategy that selectively prioritizes salient tokens while suppressing the influence of redundant ones. In the following section, we introduce VLMQ, a dedicated quantization framework for VLMs that leverages token-wise importance to counteract redundancy-induced quantization bias.

## 4 VLMQ

Considering the aforementioned limitations, we propose VLMQ, an accurate PTQ framework for VLMs that calibrates quantization in an importance-aware manner. Concretely, (1) to identify salient tokens, we introduce a diagonal gradient-derived importance factor, where each element reflects token-level importance. Then, (2) to compute this factor efficiently, we acquire raw gradients through a single lightweight block-wise backpropagation. Furthermore, (3) we then refine the optimization objective by incorporating the proposed importance factor, enabling selective emphasis on salient tokens during quantization.

### 4.1 EFFECTIVE GRADIENT-DRIVEN IMPORTANCE FACTOR

To quantify the contribution of tokens across modalities during calibration, we introduce our gradient-driven and token-level importance factors as below.

**Theorem 4.1.** *The target loss perturbation $\Delta \mathcal{L}$ can be approximated by the first-order error as*

$$\Delta \mathcal{L} \approx \Delta \theta \mathbf{p}^{(\Delta \theta), \top} + \mathcal{O}(|\theta|^2) \approx \Delta \mathbf{z} \mathbf{p}^{(\Delta \mathbf{z}), \top}, \tag{3}$$

*where $\theta \in \mathbb{R}^D$ and $\mathbf{z} \in \mathbb{R}^Q$ are the stacking weight being quantized and layer output, respectively.*

The proof is provided in Appendix D. Theorem 4.1 demonstrates that the loss perturbation $\Delta \mathcal{L}$ is influenced by two contributors: the **output errors** ($\Delta \mathbf{z}$) and the **gradients** ($\mathbf{p}^{(\Delta \mathbf{z})}$). A visualization of token-wise errors and gradients is shown in Figure 3. It is clear that the magnitudes of these two factors are not necessarily correlated. Specifically, although the errors attributed to different tokens are very similar, there is a clear disparity in their corresponding gradients. For instance, the gradients for redundant vision tokens are considerably smaller than those for important text tokens. Unlike previous studies (Lin et al., 2023; Frantar et al., 2022) that optimize quantization only by minimizing layer-wise MSE (*i.e.*, $||\Delta \mathbf{z}||^2$), in this work, we incorporate this gradient information to capture the variance of importance between tokens. This helps separately consider different modalities at the token level while downscaling the importance of redundant vision tokens during quantization. For simplicity, we omit the superscript of $\mathbf{p}^{(\Delta \mathbf{z})}$ and denote it for a certain layer as $\mathbf{P} \in \mathbb{R}^{C_o \times N}$ in the following paragraphs.

**Gradient processing.** We now describe the acquisition of the token-wise importance factor $\mathbf{G}$ (a diagonal matrix) from the raw gradients $\mathbf{P}$. Formally, the importance factor is defined as:

$$\mathbf{G} = \text{Diag}\left(\left[\overline{|\mathbf{P}|}_0, \overline{|\mathbf{P}|}_1, \ldots, \overline{|\mathbf{P}|}_{N-1}\right]\right), \text{ where } \overline{|\mathbf{P}|}_n = \frac{1}{C_o} \sum_{i=0}^{C_o-1} |\mathbf{P}|_{i,n}. \tag{4}$$

Figure 4 illustrates this process. We convert the full raw gradients into a diagonal importance factor $\mathbf{G}$. To validate the effectiveness of $\mathbf{G}$, alternative importance factors, such as attention score-style

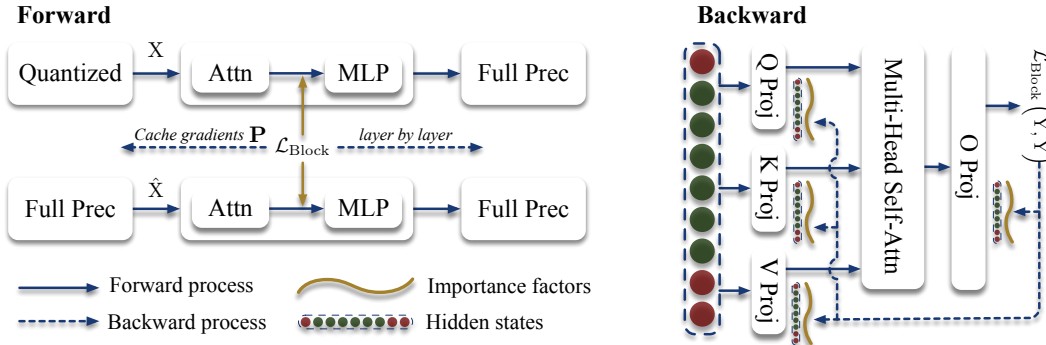

**Figure 5:** Pipeline of computing importance factors. The "Forward" module illustrates the quantization dataflow across decoding layers, where a breakpoint is set at the output of each attention module to compute the local loss $\mathcal{L}_{\text{Block}}$ and trigger a localized backward pass. The "Backward" module details the internal operations within an attention block, where gradients of each linear projection output are cached to derive token-level importance factors.

metrics (Chen et al., 2024b; Dhouib et al., 2025), have been considered. However, they fail to accurately reflect token-wise importance variance across modalities, resulting in substantial performance degradation (see Section 5.3).

## 4.2 EFFICIENT GRADIENT ACQUISITION

To extract raw gradients, we consider three potential types of target loss: *i.e.*, 1) layer-wise distillation loss, 2) block-wise distillation loss, and 3) network-wise supervised fine-tuning (SFT) loss. The layer-wise approach is relatively efficient but fails to capture cross-layer dependencies. The network-wise approach provides a more accurate gradient information, yet it incurs prohibitive computational cost and risks overfitting the quantized model to the limited calibration dataset (Gong et al., 2025). Details can be found in Section 5.3. To balance efficiency and effectiveness, we adopt a compromise solution by employing the block-wise approach.

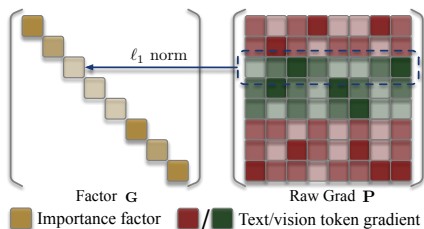

**Figure 4:** Derivation of Visualization of importance factors.

**Efficient block-wise backpropagation.** We place activation hooks immediately after attention modules to extract hidden states. This enables the computation of a localized loss $\mathcal{L}_{\text{Block}}$ between the semi-quantized model and its auxiliary full precision counterpart, which is used to trigger a one-time localized backward pass per block. Here, we define a block as an attention module as formulated in equation 1, which encompasses multi-head self-attention along with Q/K/V/O projections and the residual stream. Additionally, we employ the block MSE as the localized objective as

$$\mathcal{L}_{\text{Block}} = \left\| \text{Attn}(\mathbf{X}) - \text{Attn}(\hat{\mathbf{X}}) \right\|_2^2, \tag{5}$$

where $\mathbf{X}$ and $\hat{\mathbf{X}}$ denote the full-precision activations and the activations with preceding layers quantized, respectively. Although in theory one could define an entire decoding layer as a block, this choice typically incurs prohibitive memory overhead (Ding et al., 2023), making the approach impractical for resource-constrained hardware.

## 4.3 IMPORTANCE-AWARE OBJECTIVE

We now present our refined optimization objective incorporating the importance factor $\mathbf{G}$. In contrast to prior works (Lin et al., 2023; Frantar et al., 2022) that minimize the output MSE, we adopt an importance-aware formulation, formally defined as

$$\arg\min_{\hat{\mathbf{w}}} = ||(\Delta \mathbf{w} \mathbf{X} - \Delta \hat{\mathbf{w}} \mathbf{X}) \, \mathbf{G}||_2^2, \tag{6}$$

Note that this formulation can be seamlessly integrated into diverse PTQ frameworks. In this work, we adopt GPTAQ (Li et al., 2025b) as our precursor algorithm due to its simplicity and ease of implementation. To quantize the entire model with the proposed VLMQ, we progressively alternate between lightweight block-wise backpropagation and importance-aware calibration, thereby capturing error propagation dynamics more effectively. The detailed framework formulation and theoretical derivations are presented in Appendix C.

## 5 EXPERIMENTS

### 5.1 IMPLEMENTATION DETAILS

**Models.** We conduct experiments on open-source SOTA VLMs, including LLaVA-OneVision-0.5B/7B, Qwen2.5-VL-2B/7B/32B-Instruct, and Qwen2-VL-7B-Instruct. For the LLaVA-OneVision series, we select versions adopting Qwen2 as LM backbones and SigLIP-400M (Zhai et al., 2023) as their vision encoder.

**Calibration.** We select the improved COCO Caption dataset introduced by ShareGPT4V (Chen et al., 2024c). We randomly sample 512 text-image pairs as our calibration set. Unless otherwise specified, we use the default dampening ratio $\lambda = 0.01$ and enable act_order for all experiments. Other details can be found in Appendix E.

**Evaluation.** We undertake an extensive evaluation of quantized VLMs using multiple challenging vision-language tasks based on the LMMs-Eval framework (Zhang et al., 2024a). Specifically, we employ representative tasks including ChartQA (Masry et al., 2022), DocVQA (Mathew et al., 2021), MME-RealWorld (Zhang et al., 2024b), OCRBench (Liu et al., 2024), ScienceQA (Lu et al., 2022), SeedBench 2 Plus (Li et al., 2024b), and TextVQA (Singh et al., 2019), thereby comprehensively measuring the text recognition, visual perception, and visual reasoning capabilities of quantized models. Flash Attention (Dao et al., 2022) is enabled, and default configurations are used throughout all experiments.

### 5.2 MAIN RESULTS

**INT3g128 quantization.** We evaluate the zero-shot performance of INT3g128 (group_size=128) quantized models across eight vision-language benchmarks, as shown in Table 2. VLMQ demonstrates its overall superiority across various benchmarks. Notably, for Qwen2-VL-7B-Instruct-INT3g128, the proposed VLMQ achieves accuracies of 57.34% and 55.47% on MME-RealWorld (English/Chinese), respectively, and maintains competitive performance across other QA tasks, showcasing its robustness and versatility. Furthermore, for other model variants, VLMQ continues to narrow the gap between full-precision models and their quantized counterparts. For Qwen2-VL-2B-Instruct-INT3g128, VLMQ generally achieves improved or comparable performance across various benchmarks. However, it slightly lags behind GPTAQ, particularly due to GPTAQ's superior performance on MME-RealWorld (English), where GPTAQ outperforms the full-precision model by 3.85%. We also present the INT3 (w/o group-wise quantization) quantization comparison table in Appendix E, where VLMQ demonstrates notable performance improvements across multiple benchmarks, further highlighting its effectiveness in various settings.

**INT2g128 quantization.** We further evaluate the performance of INT2 group-wise quantized models with group_size=128 as shown in Table 3. Under these ultra-low-bit settings, transformation-based methods including AWQ (Lin et al., 2023) and MBQ (Li et al., 2025a) fail to produce desirable results. We choose GPTQ as our foundation algorithm for some experiments (see detailed explanation in Appendix E). Evidently, we observe a significant performance improvement of VLMQ under the INT2g128 configuration. For instance, the MME-RealWorld (Chinese) benchmark demonstrates a substantial 16.45% accuracy gain for Qwen2.5-VL-7B-Instruct-INT2g128 compared to GPTQ. This clear enhancement under the INT2g128 setting further validates the effectiveness of VLMQ in ultra-low-bit quantization, highlighting its ability to maintain model performance even with highly constrained bit-widths.

### 5.3 ABLATION STUDY

**Table 2:** INT3g128 performance comparison across different quantization methods on eight benchmarks, which are ChartQA, DocVQA (Validation set), MME-RealWorld (English), MME-RealWorld (Chinese), OCR-Bench, ScienceQA, SeedBench 2 Plus, and TextVQA (Validation set), respectively.

| Method | Chart | Doc$^{val}$ | MME$_{en}$ | MME$_{cn}$ | OCR | SciQA | Seed$^{2+}$ | Text$^{val}$ | Avg ($\uparrow$) |
|---|---|---|---|---|---|---|---|---|---|
| *Qwen2-VL-2B-Instruct-INT3g128* | | | | | | | | | |
| Full Prec | 72.61 | 89.35 | 40.27 | 43.59 | 76.60 | 74.23 | 61.79 | 79.38 | 67.23 |
| AWQ | 62.96 | 83.96 | 30.56 | 31.89 | 66.90 | 65.95 | 53.67 | 73.23 | 58.64 |
| MBQ | 62.80 | 82.07 | 28.96 | 33.45 | 66.30 | 67.72 | 54.98 | 73.40 | 58.71 |
| GPTQ | 62.60 | 85.29 | 37.80 | 35.54 | 70.50 | 67.15 | 58.15 | 77.23 | 61.78 |
| GPTAQ | 63.60 | 84.92 | 44.12 | 42.10 | 71.60 | 66.61 | 56.79 | 77.81 | 63.44 |
| VLMQ | 67.00 | 85.06 | 37.75 | 42.40 | 69.30 | 67.20 | 56.92 | 77.55 | 62.90 |
| *Qwen2-VL-7B-Instruct-INT3g128* | | | | | | | | | |
| Full Prec | 81.44 | 93.89 | 57.30 | 56.09 | 80.90 | 85.55 | 69.26 | 82.02 | 75.81 |
| AWQ | 77.60 | 93.46 | 56.72 | 49.97 | 77.40 | 82.39 | 66.84 | 78.91 | 72.91 |
| MBQ | 77.20 | 93.68 | 56.79 | 50.50 | 76.60 | 82.24 | 65.74 | 78.67 | 72.68 |
| GPTQ | 79.08 | 92.21 | 54.47 | 49.21 | 79.80 | 83.16 | 66.80 | 81.02 | 73.22 |
| GPTAQ | 78.92 | 92.29 | 55.83 | 52.12 | 79.00 | 83.05 | 67.28 | 80.98 | 73.68 |
| VLMQ | 79.04 | 92.45 | 57.34 | 55.47 | 79.70 | 82.32 | 67.68 | 81.21 | **74.40**$_{(+0.72\%)}$ |
| *Qwen2.5-VL-7B-Instruct-INT3g128* | | | | | | | | | |
| Full Prec | 83.20 | 94.72 | 58.55 | 52.98 | 84.40 | 88.26 | 70.62 | 83.05 | 76.97 |
| GPTQ | 77.32 | 93.91 | 58.24 | 47.39 | 82.10 | 86.23 | 71.15 | 81.79 | 74.77 |
| GPTAQ | 75.36 | 93.76 | 56.32 | 44.33 | 83.40 | 86.28 | 69.74 | 82.25 | 73.93 |
| VLMQ | 78.76 | 93.83 | 58.32 | 48.74 | 82.60 | 85.52 | 69.43 | 81.82 | **74.88**$_{(+0.11\%)}$ |
| *LLaVA-OneVision-7B-INT3g128* | | | | | | | | | |
| Full Prec | 80.08 | 87.09 | 57.39 | 53.93 | 62.10 | 89.98 | 64.82 | 75.96 | 71.42 |
| GPTQ | 78.40 | 84.53 | 55.20 | 49.64 | 60.30 | 88.85 | 63.46 | 74.55 | 69.37 |
| GPTAQ | 78.64 | 84.06 | 54.82 | 50.53 | 59.80 | 88.26 | 64.82 | 74.76 | 69.46 |
| VLMQ | 77.92 | 83.89 | 54.26 | 51.48 | 62.10 | 88.42 | 63.99 | 74.74 | **69.60**$_{(+0.14\%)}$ |

**Table 3:** INT2g128 performance comparison. $\ddagger$ indicates that we choose GPTQ as our precursor algorithm instead of GPTAQ.

| Method | Chart | Doc$^{val}$ | MME$_{en}$ | MME$_{cn}$ | OCR | SciQA | Seed$^{2+}$ | Text$^{val}$ | Avg ($\uparrow$) |
|---|---|---|---|---|---|---|---|---|---|
| *Qwen2-VL-7B-Instruct-INT2g128* | | | | | | | | | |
| AWQ | 0.00 | 0.13 | 11.58 | 7.47 | 0.60 | 2.26 | 9.18 | 0.11 | 3.92 |
| MBQ | 0.00 | 0.06 | 12.46 | 7.94 | 0.90 | 3.14 | 9.00 | 0.06 | 4.19 |
| GPTQ | 56.44 | 74.90 | 41.33 | 34.11 | 62.60 | 50.37 | 51.78 | 71.93 | 55.43 |
| GPTAQ | 56.08 | 72.57 | 37.91 | 33.80 | 61.30 | 59.70 | 51.25 | 67.98 | 55.07 |
| VLMQ $\ddagger$ | 55.32 | 75.76 | 41.97 | 35.59 | 62.50 | 62.32 | 53.80 | 74.80 | **57.76**$_{(+2.33\%)}$ |
| *LLaVA-OneVision-7B-INT2g128* | | | | | | | | | |
| GPTQ | 61.08 | 62.62 | 40.29 | 31.35 | 48.60 | 67.88 | 51.25 | 67.29 | 53.80 |
| GPTAQ | 62.12 | 62.81 | 42.59 | 32.45 | 49.90 | 66.12 | 50.94 | 67.35 | 54.29 |
| VLMQ | 62.76 | 64.82 | 38.66 | 31.38 | 51.40 | 69.04 | 53.32 | 68.20 | **54.95**$_{(+0.66\%)}$ |
| *Qwen2.5-VL-7B-Instruct-INT2g128* | | | | | | | | | |
| GPTQ | 57.72 | 78.79 | 44.94 | 13.89 | 70.80 | 55.77 | 48.40 | 74.36 | 55.58 |
| GPTAQ | 53.48 | 74.82 | 39.38 | 3.60 | 67.70 | 2.95 | 17.83 | 73.79 | 41.69 |
| VLMQ $\ddagger$ | 59.68 | 73.20 | 41.27 | 30.34 | 69.30 | 62.72 | 57.27 | 65.88 | **57.46**$_{(+1.88\%)}$ |

**Ablation on importance-aware strategies.** To validate the core idea of importance-aware PTQ, we compare four variants on Qwen2-VL-Instruct-INT2g128: GPTQ, GPTAQ, our full VLMQ, and a version without importance weighting (denoted as VLMQ$_{naive}$). In VLMQ$_{naive}$, a subset of vision tokens is randomly marked as low-importance (LI) and uniformly down-weighted (DW), and we conduct a 3×3 grid search over LI Ratio and DW Factor. As shown in Table 4, even this coarse token-weighting strategy outperforms GPTQ and GPTAQ in most settings, confirming the general effectiveness of reducing redundant visual tokens. Our full gradient-driven VLMQ achieves the highest accuracy, highlighting the necessity of fine-grained, importance-aware weighting for fully leveraging vision token redundancy.

**Table 4:** Ablation studies on importance-aware strategies.

| Method | LI Ratio | DW Factor | TextVQA | DocVQA |
|---|---|---|---|---|
| Full Prec | - | - | 82.02 | 93.89 |
| GPTQ | - | - | 71.93 | 74.90 |
| GPTAQ | - | - | 67.98 | 72.57 |
| VLMQ $_{naive}$ | | 0.01 | 70.66 | 72.12 |
| VLMQ $_{naive}$ | 0.25 | 0.05 | 72.12 | 72.63 |
| VLMQ $_{naive}$ | | 0.10 | 70.89 | 72.10 |
| VLMQ $_{naive}$ | | 0.01 | 72.38 | 72.76 |
| VLMQ $_{naive}$ | 0.50 | 0.05 | 69.83 | 71.93 |
| VLMQ $_{naive}$ | | 0.10 | 71.79 | 73.67 |
| VLMQ $_{naive}$ | | 0.01 | 69.74 | 69.44 |
| VLMQ $_{naive}$ | 0.75 | 0.05 | 68.17 | 71.05 |
| VLMQ $_{naive}$ | | 0.10 | 71.21 | 70.33 |
| VLMQ (Ours) | Grad-driven | Grad-driven | **74.80** | **75.76** |

**Ablation on importance factor type.** As discussed in Section 4, in addition to gradient information, prior works such as FastV (Chen et al., 2024b) and PACT (Dhouib et al., 2025) incorporate attention score-based factors to estimate token significance. We evaluate the quantization performance with different factor types under INT-3 quantization in Table 5. For a fair comparison, all other settings are kept

**Table 5:** Ablation study on the importance factor type. Reported results are under the INT-3 setting.

| Factor Source | Style | Avg Acc ($\uparrow$) |
|---|---|---|
| *Qwen2-VL-7B-Instruct, Avg Acc: 75.82%* | | |
| Constant | Diag($\mathbf{1}$) | 69.03% |
| Attention score | FastV (Chen et al., 2024b) | 68.29% |
| Attention score | PACT (Dhouib et al., 2025) | 67.07% |
| Gradient | VLMQ (ours) | **69.70%** |

consistent across experiments. The results demonstrate that our *formally proved* gradient-driven importance factor preserves the performance of quantized models, while *empirical* attention score-based factors lead to substantial performance degradation. Extended results can be found in Table 12.

**Ablation on backpropagation granularity.** To acquire the raw gradient, we have three potential target losses: 1) layer-wise distillation loss, 2) block-wise distillation loss, and 3) network-wise SFT loss (cross-entropy between predictions and ground truth labels). We provide the corresponding comparison in Table 6. Apparently, the block-wise manner, which is exactly used in the proposed VLMQ, provides an effective and efficient solution. We speculate the reasons behind the failure of layer-wise and network-wise manner lie in the lack of catching cross-

**Table 6:** Ablation study on backward granularity. Reported results are under the INT-3 setting.

| Loss Type | Avg Acc ($\uparrow$) | GPU Hours |
|---|---|---|
| *Qwen2-VL-7B-Instruct, Avg Acc: 75.82%* | | |
| $\mathcal{L}_{Layer} = \text{MSE}(\cdot)$ | 67.77% | 0.29 |
| $\mathcal{L}_{Network} = \text{CE}(\cdot)$ | 68.64% | 0.70 |
| $\mathcal{L}_{Block} = \text{MSE}(\cdot)$ | 69.70% | 0.21 |

layer dependency and overfitting in the limited calibration datasets. What's more, our efficiency is guaranteed by the single lightweight block-wise backpropagation, which yields the lowest quantization hours among these three. Extended results are provided in Table 13.

**Ablation on layer selection.** We also investigate the layers to be enhanced by our proposed VLMQ. As shown in Table 7, we find that placing the breakpoint at o_proj_in and skipping the enhancement of o_proj leads to failure. We speculate that breaking the integrity of a residual stream will severely damage performance. However, our block-split strategy still originates from an empirical setting like previous works (Li et al., 2021; Li & Panda, 2024; Ding et al., 2023). A more fine-grained block-split

**Table 7:** Ablation study on layers to enhance under the INT-3 setting.

| Breakpoint | Layers | Avg Acc ($\uparrow$) |
|---|---|---|
| *Qwen2-VL-7B-Instruct, Avg Acc: 75.82%* | | |
| - | $\varnothing$ | 69.03% |
| o_proj_in | {x_proj}$_{x=q/k/v}$ | 55.54% |
| attn_out | {x_proj}$_{x=q/k/v}$ | 56.21% |
| attn_out | {x_proj}$_{x=q/k/v/o}$ | **69.70%** |

strategy is required to be explored, which we leave to our future work. Extended results are provided in Table 14.

## 5.4 GENERALIZATION

**Table 8:** Performance comparison on reasoning and language-heavy benchmarks.

| Method | MMMU$^{\text{val}}$ (%) | HellaSwag (%) | TextCaps (CIDEr) |
|---|---|---|---|
| Full Prec | 50.44 | 77.63 | 152.00 |
| GPTQ | 32.22 | 51.29 | 55.06 |
| GPTAQ | 22.25 | 49.82 | 46.52 |
| VLMQ | **33.67** | **53.19** | **55.90** |

**Table 9:** Memory usage and quantization latency of different model sizes on a single H100 80GB GPU.

| Model Size | Metric | GPU Cost | | |
|---|---|---|---|---|
| | | GPTQ | GPTAQ | VLMQ |
| 2B | Peak Mem (GB) | 4.99 | 10.59 | 12.78$_{(+2.19\,\text{GB})}$ |
| | Time (Hour) | 0.10 | 0.11 | 0.13$_{(+1.2\,\text{mins})}$ |
| 7B | Peak Mem (GB) | 17.37 | 24.76 | 29.05$_{(+4.29\,\text{GB})}$ |
| | Time (Hour) | 0.21 | 0.27 | 0.29$_{(+1.8\,\text{mins})}$ |
| 32B | Peak Mem (GB) | 18.99 | 38.94 | 41.56$_{(+2.62\,\text{GB})}$ |
| | Time (Hour) | 0.86 | 1.08 | 1.21$_{(+6.0\,\text{mins})}$ |

To assess performance on language-heavy and multimodal reasoning tasks, we evaluate VLMQ on HellaSwag (Zellers et al., 2019), TextCaps (Zellers et al., 2019), and MMMU (Yue et al., 2023), respectively. These benchmarks cover pure language understanding, visually grounded language generation, and broad-domain multimodal reasoning, allowing us to evaluate the generalization ability of VLMQ beyond standard VQA-style tasks. These benchmarks cover pure language understanding, visually grounded language generation, and broad-domain multimodal reasoning, allowing us to evaluate the generalization ability of VLMQ beyond standard VQA-style tasks.

## 5.5 QUANTIZATION EFFICIENCY

Table 9 reports the quantization latency and peak memory consumption of VLMQ compared with GPTQ and GPTAQ across different model scales. Since VLMQ preserves the original GPTQ-format quantization scheme, it maintains compatibility with existing efficiency optimizations and introduces only token-level importance estimation as an additional component. As a result, the computational overhead remains minimal, reflected by only minor latency increases of <10 minutes depending on model size. In terms of memory usage, the additional cost originates primarily from storing the activation gradients required for one local backward pass per block. This overhead is modest relative to the total memory budget of modern accelerators and remains well within the capacity of a single H100 80GB GPU even for 32B-scale models. Overall, the results confirm that VLMQ provides its performance gains with only lightweight increases in memory and negligible impact on quantization time.

## 6 CONCLUSION AND LIMITATION

We introduced VLMQ, a novel importance-aware PTQ framework for VLMs. We identified the vision over-representation that hinders the direct application of existing LLM PTQ methods to VLMs. Motivated by this insight, VLMQ explicitly leverages the enhanced Hessian yielded from the importance-aware objective. VLMQ effectively enhances the performance of quantized VLMs across various vision-language benchmarks, making large VLMs more practical for deployment under ultra-low-bit settings. In terms of limitations, our evaluation primarily focuses on image-text tasks. However, we believe VLMQ can generalize to broader applications, such as video understanding and language-only tasks, which we leave for future exploration.

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

APPENDIX

OUTLINE

We provide additional details in the appendix, organized as follows

- Section A supplements related works on Hessian-based quantization for LLMs. In particular, we summarize the motivation, methodology, and limitations of existing approaches, and provide an overview of GPTAQ (Li et al., 2025b) as the baseline precursor algorithm adopted in this work.
- Section B describes the detailed architectures of representative VLMs and outlines the necessary quantization preliminaries.
- Section C elaborates on the proposed VLMQ framework. We provide a step-by-step derivation of the optimization objective and the role of importance factors. For reproducibility, we also include detailed pseudo-code illustrating the complete quantization pipeline.
- Section D presents the theoretical justification of our method, including the full proof of the theorem introduced in the main paper.
- Section E reports extended experimental settings and comprehensive results. This includes dataset descriptions, evaluation metrics, implementation details, and additional evaluation results that complement the main results and further validate the effectiveness of our approach.
- Section F describes the use of LLM.

## A  RELATED WORKS

**Hessian-based LLM quantization.** Hessian information is widely used as a calibration guide to compensate for quantization errors. OBQ (Frantar & Alistarh, 2022) calibrates the quantization of linear layers in models by iteratively 1) quantizing entries that introduce minimal loss, and 2) updating the remaining unquantized ones with the guidance of Hessian $\mathbf{H} = \mathbf{X}\mathbf{X}^\top$. GPTQ (Frantar et al., 2022) extends this paradigm by formulating fixed-order entry quantization in a parallel manner, lazy block updating, and Cholesky reformulation to achieve efficient calibration. GPTAQ (Li et al., 2025b) further extends GPTQ by optimizing an asymmetric objective:

$$\arg \min_{\hat{\mathbf{w}}} = ||\Delta\mathbf{w}\mathbf{X} - \mathbf{r}||_2^2, \tag{7}$$

where $\mathbf{r} = \mathbf{w}\mathbf{X} - \mathbf{w}\hat{\mathbf{X}}$ denotes the activation residual and $\Delta\mathbf{w} = \hat{\mathbf{w}} - \mathbf{w}$ is the weight perturbation. The optimal solution is derived via Lagrangian formulation:

$$\Delta\mathbf{w} = \frac{(\hat{\mathbf{w}}_q - \mathbf{w}_q)}{\mathbf{H}_{qq}^{-1}} \cdot \mathbf{H}_{q:,}^{-1} + \mathbf{r}\mathbf{X}^\top \mathbf{H}_{-q}^{-1}, \tag{8}$$

where $q$ denotes the index of the quantized weight element. GPTAQ thereby achieves accurate quantization while better preserving model functionality, with efficiency ensured by techniques such as efficient residual decomposition. Other works (Kim et al., 2024; Edalati et al., 2025; Guan et al., 2024) focus on obtaining accurate Hessian information to enhance PTQ, enable mixed-precision quantization, and support related techniques.

## B  VLM QUANTIZATION BACKGROUND

### B.1  VLM ARCHITECTURE

As shown in Figure 6, the VLMs comprise three key components: a visual encoder, a vision-text projector, and a Language Model (LM) backbone. The visual encoders can be a pre-trained image encoder like a CLIP-style encoder (Radford et al., 2021), which is responsible for converting image/video input into vision tokens. Then they are fed into the MLP-based projector to align it with the LM embedding space. Together with encoded text tokens, they are further fed into the LM backbone. The LM backbone usually comprises multiple stacked decoding layers followed by an LM prediction head. The modality gap across hidden states in the word embedding space can be observed by PCA.

Given the multi-modal input $\mathbf{X} \in \mathbb{R}^{C_i \times N}$ of the language model backbone, we can disassemble it based on different functionalities as

$$\mathbf{X} = \mathbf{X}_{\text{sys}} \oplus \mathbf{X}_{\text{img}} \oplus \mathbf{X}_{\text{ins}} \oplus \mathbf{X}_{\text{ans}}, \qquad (9)$$

where $\mathbf{X}_{\text{sys}}$, $\mathbf{X}_{\text{img}}$, $\mathbf{X}_{\text{ins}}$, and $\mathbf{X}_{\text{ans}}$ represent system prompt, vision, user instruction, and model response tokens, respectively. $\oplus$ signifies the concatenation operation. Without loss of generality, we can categorize tokens in different modalities and rewrite equation 9 as

$$\mathbf{X} = \mathbf{X}_t \oplus \mathbf{X}_v, \qquad (10)$$

where text and vision token set $\mathbf{X}_t$ and $\mathbf{X}_v$ are $\mathbf{X}_t = \mathcal{S}\{\mathbf{X}_{\text{sys}}, \mathbf{X}_{\text{ins}}, \mathbf{X}_{\text{ans}}\}$ and $\mathbf{X}_v = \mathcal{S}\{\mathbf{X}_{\text{img}}\}$, respectively.

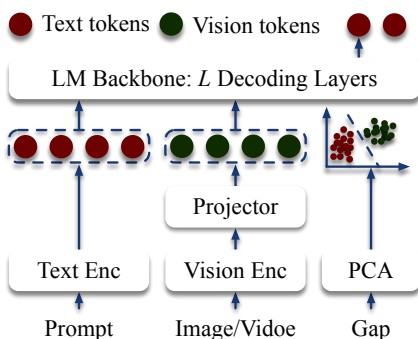

**Figure 6:** VLM architecture.

## B.2 Quantization Preliminary

This paper focuses on uniform affine quantization. Quantization essentially maps high-precision (*e.g.*, FP16/BF16) weights/activations to low-precision (*e.g.*, INT8/INT4) formats to reduce memory footprint and achieve inference acceleration, which is given by

$$\text{Quant: } \mathbf{W}_{\text{int}} = \text{Clamp}\left(\left\lfloor \frac{\mathbf{W}}{s} \right\rceil + z, \, 0, \, 2^B - 1\right), \qquad (11)$$

$$\text{Dequant: } \hat{\mathbf{W}} = s \times (\mathbf{W}_{\text{int}} - z), \qquad (12)$$

where $\mathbf{W}_{\text{int}}$ is in the INT-$B$ format and $\hat{\mathbf{W}}$ is the pseudo-quantized weight in full precision. The quantization parameters $s$ and $z$ are the quantization step size and zero-point, given by

$$s = \frac{\max(\mathbf{W}) - \min(\mathbf{W})}{2^B - 1} \qquad (13)$$

$$z = -\left\lfloor \frac{\min(\mathbf{W})}{s} \right\rceil. \qquad (14)$$

## C VLMQ Framework

### C.1 Refined Objective

In this paper, we choose the advanced GPTAQ as our precursor algorithm. Although equation 8 yields the optimal quantized weight $\hat{\mathbf{w}}$ by minimizing the squared error, it inherently assumes equal contribution across all tokens in $\mathbf{Z} = \mathbf{W}\mathbf{X}$. Such equal treatment may introduce bias due to the vision over-representation. To address this limitation, we introduce a token-level importance weighting matrix $\mathbf{G} \in \mathbb{R}^{N \times N}$, where $\mathbf{G}$ is diagonal with the $i$-th diagonal element representing the importance assigned to output token $\mathbf{Z}_{:,i}$. Incorporating importance factors into the objective leads to a refined formulation:

$$\arg\min_{\hat{\mathbf{w}}} = ||(\Delta\mathbf{w}\mathbf{X} - \mathbf{r})\mathbf{G}||_2^2,$$
$$\text{s.t. } \Delta\mathbf{w}\mathbf{e}_q^\top + \mathbf{w}_q - \hat{\mathbf{w}}_q = 0 \qquad (15)$$

where $\mathbf{e}_q$ is the one-hot vector with $q$-th element being 1. We formulate its Lagrangian as:

$$L = ||(\Delta\mathbf{w}\mathbf{X} - \mathbf{r})\mathbf{G}||_2^2 + \lambda\left(\Delta\mathbf{w}\mathbf{e}_q^\top + \mathbf{w}_q - \hat{\mathbf{w}}_q\right), \qquad (16)$$

To identify the local minima of the Lagrangian formulation, we differentiate with respect to $\Delta\mathbf{w}$ and the Lagrange multiplier $\lambda$, and solve the resulting equations by setting them to zero:

$$\begin{cases} \frac{\partial L}{\partial \Delta\mathbf{w}} = 2\Delta\mathbf{w}\widetilde{\mathbf{H}} - 2\widetilde{\mathbf{r}}\widetilde{\mathbf{X}}^\top + \lambda\mathbf{e}_q = 0, \\ \frac{\partial L}{\partial \lambda} = \Delta\mathbf{w}\mathbf{e}_q^\top + \mathbf{w}_q - \hat{\mathbf{w}}_q = 0, \end{cases} \qquad (17)$$

which yields the solution for the first equation as

$$\Delta\mathbf{w}\widetilde{\mathbf{H}} = -\lambda/2\mathbf{e}_q + \widetilde{\mathbf{r}}\widetilde{\mathbf{X}}^\top. \qquad (18)$$

---

**Algorithm 1** VLMQ for one decoding layer

---

**Input**: Calibration input $\hat{\mathbf{X}}$, FP input $\mathbf{X}$, and decoding layer $\mathcal{W} \leftarrow \{\texttt{Attn}(\cdot), \texttt{MLP}(\cdot)\}$.

    Forward $\texttt{Attn}(\mathbf{X})$ and $\texttt{Attn}(\hat{\mathbf{X}})$.
    Cache $\mathbf{G} \leftarrow \texttt{Backward}(\mathcal{L}_{\text{Block}})$ based on Equation 5.
    **for** $\mathbf{W}_{\text{name}} \in \mathcal{W}$ **do**
        **if** name $\in \{\texttt{q\_proj}, \texttt{k\_proj}, \texttt{v\_proj}, \texttt{o\_proj}\}$ **then**
            Compute $\widetilde{\mathbf{H}}$, $\tilde{\mathbf{r}}$, and $\widetilde{\mathbf{X}}$ based on Equation 19.
            $\hat{\mathbf{W}}_{\text{name}} \leftarrow \text{VLMQ}\left(\mathbf{W}_{\text{name}}, \widetilde{\mathbf{H}}, \tilde{\mathbf{r}}, \widetilde{\mathbf{X}}\right)$
        **else**
            Compute normal $\mathbf{H}$ and $\mathbf{r}$.
            $\hat{\mathbf{W}}_{\text{name}} \leftarrow \text{GPTAQ}\left(\mathbf{W}_{\text{name}}, \mathbf{H}, \mathbf{r}, \mathbf{X}\right)$
        **end if**
    **end for**

---

The following derivation closely follows the formal framework established in GPTAQ (Li et al., 2025b), with the key distinction being the substitution of the vanilla Hessian, residual, and activation matrix by their importance-aware counterparts, as defined in equation 19. We present the corresponding optimal weight updating in VLMQ as

$$\Delta\mathbf{w} = \frac{(\hat{\mathbf{w}}_q - \mathbf{w}_q)}{\left[\mathbf{X}\mathbf{G}^2\mathbf{X}^\top\right]_{qq}^{-1}} \cdot \widetilde{\mathbf{H}}_{q:,}^{-1} + \mathbf{r}\mathbf{G}^2\mathbf{X}^\top\left[\mathbf{X}\mathbf{G}^2\mathbf{X}^\top\right]_{-q}^{-1}$$

$$= \frac{(\hat{\mathbf{w}}_q - \mathbf{w}_q)}{\widetilde{\mathbf{H}}_{qq}^{-1}} \cdot \widetilde{\mathbf{H}}_{q:,}^{-1} + \tilde{\mathbf{r}}\widetilde{\mathbf{X}}^\top\widetilde{\mathbf{H}}_{-q}^{-1}, \tag{19}$$

where $\widetilde{\mathbf{H}} = \mathbf{X}\mathbf{G}^2\mathbf{X}^\top$, $\tilde{\mathbf{r}} = \mathbf{r}\mathbf{G}$, and $\widetilde{\mathbf{X}} = \mathbf{X}\mathbf{G}$ indicate the importance-aware Hessian, residual, and activation matrix yielded from the refined objective (equation 15). Our importance-aware weight update presented in equation 19 remains formally consistent with the original GPTAQ (Li et al., 2025b) formulation. Distinct from BoA (Kim et al., 2024), which attributes importance at the channel level, our approach assigns importance scores at the token-level activation granularity. This design is orthogonal to the parallelization strategies proposed in GPTQ (Frantar et al., 2022) and GPTAQ (Li et al., 2025b). The compatibility allows us to directly inherit efficiency tricks (*e.g.*, Cholesky reformulation) proposed therein.

## C.2 VLMQ

The detailed algorithm is provided in Algorithm 1. For each decoding layer, we begin by caching the quantized input $\mathbf{X}$ and its full-precision counterpart $\hat{\mathbf{X}}$. Prior to quantizing the linear modules, we perform a block-wise forward and backward pass to obtain gradients of the Q/K/V/O projection outputs. The token-level importance factors are then computed from these gradients, as described in equation 4. For quantizing the Q/K/V/O projections, we construct the importance-aware Hessian and apply the weight update rule defined in equation 19. For the Up/Gate/Down projections, we formulate the vanilla Hessian and similarly apply the weight update rule in equation 8.

# D THEORETICAL PROOF

## D.1 PROOF OF THEOREM 4.1

Theorem 4.1 establishes a connection between block-wise loss perturbation and layer output error. The proof begins with a Taylor expansion of the block-wise MSE loss with respect to the quantization noise. Specifically, the perturbation in the loss induced by quantized weights can be approximated as

$$\Delta\mathcal{L}_{\text{Block}} = \mathcal{L}_{\text{Block}}\left(\theta + \Delta\theta\right) - \mathcal{L}_{\text{Block}}\left(\theta\right) \tag{20}$$

$$= \Delta\theta\mathbf{p}^{(\Delta\theta),\top} + \mathcal{O}(|\theta|^2), \tag{21}$$

where $\theta \in \mathbb{R}^D$ is the stacking weight being quantized. By defining the layer output as $\mathbf{z} \in \mathbb{R}^Q$, we further derive the equation 20 as

$$\Delta \mathcal{L}_{\text{Block}} \approx \Delta \theta \mathbf{p}^{(\Delta \theta), \top} \tag{22}$$

$$= \sum_{i=1}^{D} \Delta \theta_i \frac{\partial \mathcal{L}_{\text{Block}}}{\partial \theta_i} \tag{23}$$

$$= \sum_{i=1}^{D} \Delta \theta_i \left( \sum_{j=1}^{Q} \frac{\partial \mathcal{L}_{\text{Block}}}{\partial z_j} \frac{\partial z_j}{\partial \theta_i} \right) \tag{24}$$

$$= \sum_{j=1}^{Q} \frac{\partial \mathcal{L}_{\text{Block}}}{\partial z_j} \sum_{i=1}^{D} \left( \Delta \theta_i \frac{\partial z_j}{\partial \theta_i} \right) \tag{25}$$

$$= \sum_{j=1}^{Q} \Delta z_j \frac{\partial \mathcal{L}_{\text{Block}}}{\partial z_j} \tag{26}$$

$$= \Delta \mathbf{z} \mathbf{p}^{(\Delta \mathbf{z}), \top}. \tag{27}$$

Proof done.

# E EXPERIMENTS

## E.1 IMPLEMENTATION DETAILS

**Calibration.** We randomly sample 512 data samples from COCO Caption datasets (Chen et al., 2024c) as the calibration set. Each sample is structured according to equation 9 and concatenated with the default prompt template. The input sequence length is adjusted based on the target model family: for Qwen2-VL and Qwen2.5-VL series, we truncate inputs to 512 tokens, while for LLaVA-OneVision series, the length is set to 986 tokens. To ensure semantic completeness, we discard samples whose vision token segments are truncated mid-way, as such instances result in incomplete symbolism. For quantization configurations, we adopt most settings from GPTAQ (Li et al., 2025b). Specifically, we enable `act_order` to enhance numerical stability. For group-wise quantization, we disable `static_group` to allow dynamic group-wise quantization parameter assignment.

**Evaluation.** We implement VLMQ using the Huggingface Transformers library (Wolf et al., 2019) on top of the PyTorch framework. For evaluation, we leverage the open-source LMMs-Eval toolkit (Zhang et al., 2024a), with minor adaptations tailored to our quantized settings. During evaluation, we observe that the default answer matching mechanism in LMMs-Eval is suboptimal for ScienceQA, especially under ultra-low-bit quantization scenarios. In particular, although the model often provides the correct answer, it may not strictly adhere to the required output format, namely, returning only the option letter. For instance, a response such as "The answer is B." semantically matches the correct answer but fails to conform to the expected literal format (e.g., "B" or "B."), leading to a false negative in accuracy computation. To mitigate this mismatch, we introduce a robust post-processing pipeline for automatic answer normalization and extraction. This process involves parsing the model's textual response and identifying the most probable option letter based on regular expressions and contextual cues.

## E.2 ADDITIONAL RESULTS.

**Additional illustration on INT2g128 quantization.** While taking GPTAQ as our precursor algorithm generally yields satisfactory results under INT3g128 quantization, applying this configuration to INT2g128 leads to severe performance degradation. In particular, GPTAQ produces even worse results than GPTQ under this ultra-low-bit setting. Taking Qwen2.5-VL-7B-Instruct-INT2g128 as an example, GPTAQ achieves only 41.69% accuracy, significantly lower than the 55.58% achieved by GPTQ. Additionally, the Qwen2.5-VL-7B-Instruct-INT2g128 fails on MME-RealWorld (Chinese) and ScienceQA benchmarks, achieving only 3.60% and 2.95% accuracy, respectively. **A plausible explanation is that, under extremely low-bit quantization, the residual error in GPTAQ**

**Table 10:** INT3 quantization performance comparison across different quantization methods on eight benchmarks. $\dagger$ indicates that we use the token-level $\ell_2$-norm of row gradients instead of $\ell_1$-norm as our importance factors.

| Method | Chart | Doc[val] | MME[en] | MME[cn] | OCR | SciQA | Seed[2+] | Text[val] | Avg ($\uparrow$) |
|---|---|---|---|---|---|---|---|---|---|
| *Qwen2-VL-2B-Instruct-INT3* | | | | | | | | | |
| Full Prec | 72.61 | 89.35 | 40.27 | 43.59 | 76.60 | 74.23 | 61.79 | 79.38 | 67.23 |
| AWQ | 5.36 | 10.02 | 9.65 | 8.48 | 5.40 | 1.37 | 18.53 | 4.37 | 7.90 |
| MBQ | 3.56 | 10.19 | 10.22 | 7.79 | 4.40 | 1.44 | 8.12 | 3.90 | 6.20 |
| GPTQ | 58.84 | 74.55 | 33.25 | 30.83 | 61.50 | 65.13 | 50.37 | 73.49 | 56.00 |
| GPTAQ | 58.76 | 74.03 | 38.71 | 28.71 | 61.10 | 64.25 | 52.17 | 73.18 | 56.36 |
| VLMQ | 59.44 | 76.59 | 37.94 | 32.58 | 61.90 | 63.05 | 52.92 | 74.54 | **57.37**$_{(+1.01\%)}$ |
| *Qwen2-VL-7B-Instruct-INT3* | | | | | | | | | |
| Full Prec | 81.44 | 93.89 | 57.30 | 56.09 | 80.90 | 85.55 | 69.26 | 82.02 | 75.81 |
| AWQ | 18.20 | 27.67 | 30.20 | 20.03 | 8.90 | 49.89 | 44.88 | 28.23 | 28.50 |
| MBQ | 16.52 | 24.75 | 27.23 | 21.35 | 8.10 | 35.25 | 40.10 | 27.79 | 25.13 |
| GPTQ | 72.80 | 88.35 | 47.76 | 27.90 | 72.40 | 78.26 | 63.02 | 78.28 | 66.10 |
| GPTAQ | 72.76 | 88.09 | 49.76 | 43.96 | 73.80 | 80.41 | 64.21 | 79.23 | 69.03 |
| VLMQ | 72.76 | 88.90 | 50.17 | 47.25 | 73.50 | 80.05 | 65.52 | 79.41 | **69.70**$_{(+0.67\%)}$ |
| *LLaVA-OneVision-7B-INT3* | | | | | | | | | |
| Full Prec | 80.08 | 87.09 | 57.39 | 53.93 | 62.10 | 89.98 | 64.82 | 75.96 | 71.42 |
| GPTQ | 74.24 | 77.82 | 46.47 | 42.59 | 56.60 | 85.26 | 60.65 | 71.90 | 64.44 |
| GPTAQ | 75.72 | 78.07 | 46.48 | 42.94 | 58.30 | 85.97 | 61.84 | 72.69 | 65.25 |
| VLMQ | 75.60 | 78.32 | 48.93 | 45.41 | 58.00 | 85.29 | 60.91 | 73.09 | **65.69**$_{(+0.44\%)}$ |
| VLMQ $\dagger$ | 74.56 | 78.21 | 49.12 | 45.94 | 59.20 | 86.37 | 61.05 | 73.66 | **66.01**$_{(+0.76\%)}$ |
| *Qwen2.5-VL-7B-Instruct-INT3* | | | | | | | | | |
| Full Prec | 83.20 | 94.72 | 58.55 | 52.98 | 84.40 | 88.26 | 70.62 | 83.05 | 76.97 |
| GPTQ | 62.44 | 89.33 | 42.31 | 27.33 | 77.70 | 79.39 | 66.58 | 78.38 | 65.43 |
| GPTAQ | 63.68 | 90.41 | 47.18 | 36.49 | 79.60 | 80.52 | 66.40 | 77.94 | 67.78 |
| VLMQ | 65.32 | 91.36 | 45.50 | 37.89 | 80.00 | 81.70 | 67.32 | 79.28 | **68.55**$_{(+0.77\%)}$ |
| *Qwen2.5-VL-32B-Instruct-INT3* | | | | | | | | | |
| Full Prec | 69.24 | 92.51 | 60.21 | 60.37 | 80.30 | 92.69 | 71.67 | 77.06 | 75.51 |
| GPTQ | 43.88 | 88.26 | 51.77 | 47.76 | 76.50 | 87.22 | 64.87 | 73.88 | 74.20 |
| GPTAQ | 52.52 | 88.72 | 52.13 | 42.67 | 77.40 | 79.46 | 67.89 | 72.88 | 73.68 |
| VLMQ | 49.84 | 87.47 | 53.05 | 43.45 | 76.90 | 83.92 | 65.96 | 73.44 | **74.31**$_{(+0.11\%)}$ |
| *LLaVA-OneVision-0.5B-INT3* | | | | | | | | | |
| Full Prec | 61.48 | 69.01 | 38.94 | 32.13 | 57.60 | 63.36 | 53.05 | 65.83 | 63.46 |
| GPTQ | 9.32 | 9.35 | 13.40 | 21.63 | 21.70 | 10.09 | 0.57 | 4.34 | 10.96 |
| GPTAQ | 2.88 | 7.70 | 9.20 | 11.27 | 20.70 | 5.35 | 1.23 | 9.22 | 9.17 |
| VLMQ $\ddagger$ | 8.04 | 16.40 | 20.42 | 15.08 | 29.20 | 7.90 | 3.03 | 12.77 | **14.86**$_{(+3.90\%)}$ |

**Table 11:** INT4 quantization performance comparison across different quantization methods on eight benchmarks.

| Method | Chart | Doc[val] | MME[en] | MME[cn] | OCR | SciQA | Seed[2+] | Text[val] | Avg ($\uparrow$) |
|---|---|---|---|---|---|---|---|---|---|
| *Qwen2-VL-7B-Instruct-INT4* | | | | | | | | | |
| GPTQ | 80.44 | 93.16 | 55.44 | 44.03 | 79.70 | 84.08 | 67.63 | 81.55 | 73.25 |
| GPTAQ | 79.88 | 93.04 | 55.55 | 47.30 | 79.50 | 84.23 | 67.85 | 81.68 | 73.63 |
| VLMQ | 80.28 | 93.23 | 55.88 | 47.73 | 80.30 | 84.30 | 67.72 | 81.48 | 73.87 |

**propagated from previously quantized layers may be continuously accumulated and significantly amplified.** This accumulated noise severely interferes with the dominant weight update, thereby compromising the undesirable quantization quality. Therefore, in cases where GPTAQ underperforms GPTQ, we opt to use GPTQ as our precursor algorithm. It is important to note that, since our proposed VLMQ framework is designed as a plug-and-play solution for VLMs, it allows for seamless adaptation from a GPTAQ-based implementation to one based on GPTQ.

**Results of INT3/INT4 quantization.** We assess the zero-shot performance of INT3/INT4 quantized models across eight vision-language benchmarks in Table 10 and Table 11. For INT3 quantization, VLMQ demonstrates overall superiority compared to baseline approaches and achieves up to 1.01% average accuracy improvement. For Qwen2.5-VL-7B-Instruct-INT3, the proposed VLMQ outper-

**Table 12:** Ablation study on the importance factor type. Reported results are under the INT-3 setting.

| Label | Chart | Doc$^{val}$ | MME$_{en}$ | MME$_{cn}$ | OCR | SciQA | Seed$^{2+}$ | Text$^{val}$ | Avg ($\uparrow$) |
|---|---|---|---|---|---|---|---|---|---|
| | *Qwen2-VL-7B-Instruct, Avg Acc: 75.82%* | | | | | | | | |
| ① | 72.76 | 88.09 | 49.76 | 43.96 | 73.80 | 80.41 | 64.21 | 79.23 | 69.03 |
| ② | 69.40 | 88.32 | 49.88 | 45.28 | 72.90 | 78.54 | 63.64 | 78.33 | 68.29 |
| ③ | 71.60 | 86.83 | 48.37 | 40.76 | 70.20 | 78.90 | 63.20 | 76.71 | 67.07 |
| ④ | 72.76 | 88.90 | 50.17 | 47.25 | 73.50 | 80.05 | 65.52 | 79.41 | 69.70 |

**Table 13:** Ablation study on backward granularity. Reported results are under the INT-3 setting.

| Label | Chart | Doc$^{val}$ | MME$_{en}$ | MME$_{cn}$ | OCR | SciQA | Seed$^{2+}$ | Text$^{val}$ | Avg ($\uparrow$) |
|---|---|---|---|---|---|---|---|---|---|
| | *Qwen2-VL-7B-Instruct, Avg Acc: 75.82%* | | | | | | | | |
| ① | 71.08 | 87.18 | 48.23 | 38.57 | 73.70 | 79.81 | 64.43 | 79.14 | 67.77 |
| ② | 69.80 | 87.72 | 49.35 | 45.17 | 73.70 | 80.62 | 64.21 | 78.54 | 68.64 |
| ③ | 72.76 | 88.90 | 50.17 | 47.25 | 73.50 | 80.05 | 65.52 | 79.41 | 69.70 |

forms its counterparts quantized by GPTQ and GPTAQ on all benchmarks except MME-RealWorld (English). Remarkably, it yields accuracy improvements of 1.64%, 1.40%, and 1.18% on ChartQA, MME-RealWorld (Chinese), and ScienceQA, respectively. For the LLaVA-OneVision-7B-INT3 model, we apply the token-level $\ell_2$-norm instead of the $\ell_1$-norm adopted in the other experiments, which we empirically find beneficial for performance enhancement. However, despite its effectiveness, the 0.5B model still exhibits a notable performance gap compared to its full-precision counterpart, highlighting the challenge of quantizing smaller VLMs under ultra-low-bit settings. We leave it as our future work. For the INT4 quantized model, VLMQ also delivers solid performance, showing small but consistent improvements over existing baselines. This confirms that the method remains effective even under moderate-bit quantization.

**Extended ablation studies.** We provide the detailed results of Table 5, Table 6, and Table 7 in Table 12, Table 13, and Table 14, respectively. The label corresponds to the line number in the compact table.

**Ablation on precursor algorithms.** To better understand the role of precursor PTQ algorithms, we compare VLMQ when applied on top of GPTQ and GPTAQ under the INT2g128 setting (Table 15). GPTQ adopts layer-isolated calibration, whereas GPTAQ performs sequential layer-wise calibration using quantized inputs, which is beneficial at moderate bit-widths (e.g., INT3) but becomes unstable under ultra–low-bit settings due to amplified accumulated error. As a result, GPTAQ exhibits degraded performance at INT2, while GPTQ remains more reliable. Across both 7B and 7B-2.5 variants, VLMQ consistently improves each precursor by up to 10% on GPTAQ, demonstrating that our importance-aware mechanism is orthogonal to the calibration pipeline and enhances both symmetric and asymmetric PTQ formulations.

### E.3 COMPATIBILITY WITH HARDWARE-OPTIMIZED KERNELS

Regarding deployment efficiency on hardware, we stress that VLMQ is fully compatible with GPTQ's quantization format. As a result, it can be seamlessly incorporated into existing works and directly utilize all hardware-optimized kernels and infrastructures designed for GPTQ, without introducing extra inference overhead. This compatibility further allows VLMQ to immediately take advantage of specialized kernels such as Marlin (Frantar et al., 2024) and ExLLaMA (Contributors, 2024), eliminating the need for additional hardware optimizations.

## F THE USE OF LLMs

The LLMs are solely used for language polishing, without involvement in technical content.

**Table 14:** Ablation study on layers to enhance under the INT-3 setting.

| Label | Chart | Doc$^{val}$ | MME$_{en}$ | MME$_{cn}$ | OCR | SciQA | Seed$^{2+}$ | Text$^{val}$ | Avg ($\uparrow$) |
|---|---|---|---|---|---|---|---|---|---|
| | | | | *Qwen2-VL-7B-Instruct, Avg Acc: 75.82%* | | | | | |
| ① | 72.76 | 88.09 | 49.76 | 43.96 | 73.80 | 80.41 | 64.21 | 79.23 | 69.03 |
| ② | 58.16 | 73.21 | 33.51 | 29.15 | 62.80 | 61.71 | 52.88 | 72.88 | 55.54 |
| ③ | 53.68 | 74.22 | 37.96 | 35.42 | 61.80 | 62.53 | 50.42 | 73.62 | 56.21 |
| ④ | 72.76 | 88.90 | 50.17 | 47.25 | 73.50 | 80.05 | 65.52 | 79.41 | 69.70 |

**Table 15:** Average accuracy improvements brought by VLMQ on INT2g128 quantized models.

| Method | Avg ($\uparrow$) |
|---|---|
| *Qwen2-VL-7B-Instruct-INT2g128* | |
| GPTQ | 55.43 |
| GPTQ + VLMQ | **57.76** |
| GPTAQ | 55.07 |
| GPTAQ + VLMQ | **56.51** |
| *Qwen2.5-VL-7B-Instruct-INT2g128* | |
| GPTQ | 55.58 |
| GPTQ + VLMQ | **57.46** |
| GPTAQ | 41.69 |
| GPTAQ + VLMQ | **52.10** |

