# OpenReview forum: "Towards Efficient Post-Training Quantization For Large Vision-Language Models Via Token-Wise Redundancy Elimination"
_ICLR.cc/2026/Conference — Submitted to ICLR 2026_

### Official Review · Reviewer_x6UG · 2025-10-31

**Soundness:** 3
**Presentation:** 3
**Contribution:** 3
**Rating:** 6
**Confidence:** 3

**Summary:**

This paper identifies two issues in VLM activations that hinder quantization: (1) visual over-representation, vision modality inputs produce an excessive number of tokens, many of which are redundant; and (2) modality gap, there is a clear separation between vision and text token distributions in latent space. To address these issues, the authors propose VLMQ, which introduces a gradient-driven token importance factor G to automatically re-weight each token’s importance, enabling the model to allocate its limited quantization precision preferentially to the truly important tokens.

**Strengths:**

A substantive assessment of the strengths of the paper, touching on each of the following dimensions: originality, quality, clarity, and significance. We encourage reviewers to be broad in their definitions of originality and significance. For example, originality may arise from a new definition or problem formulation, creative combinations of existing ideas, application to a new domain, or removing limitations from prior results.

1.The paper is well-motivated with a clear problem definition and justification of its importance. The authors effectively highlight the redundancy issue in visual tokens and the challenges of post-training quantization in vision-language models.

2.The visualizations are intuitive and help readers understand the key ideas, such as token-wise importance and the gradient-based weighting mechanism.

3.The experimental section especially the ablation studies are comprehensive, which demonstrates the effectiveness and validity of the proposed method.

**Weaknesses:**

A substantive assessment of the weaknesses of the paper. Focus on constructive and actionable insights on how the work could improve towards its stated goals. Be specific, avoid generic remarks. For example, if you believe the contribution lacks novelty, provide references and an explanation as evidence; if you believe experiments are insufficient, explain why and exactly what is missing, etc.

1.The performance gains at higher bit-widths (e.g., INT3) are relatively small — sometimes less than 0.2%, and even slightly inferior to baselines on specific models such as Qwen2-VL-2B-INT3g128.

2.The method introduces additional computational and memory overhead due to the need for block-wise backpropagation, which limits its efficiency compared to simpler post-training quantization approaches like GPTQ.

3.It is not clear how the method performs for language-heavy/language-only task. Given that GPTQ is relatively outdated for language tasks, it would be helpful to examine how the proposed token-importance weighting impacts language performance.

**Questions:**

Please list up and carefully describe any questions and suggestions for the authors. Think of the things where a response from the author can change your opinion, clarify a confusion or address a limitation. This is important for a productive rebuttal and discussion phase with the authors.

1.Include experiments on larger models such as LLaVA-13B or InternVL-26B to verify scalability.

2.Provide results for INT4 quantization, which may offer a better trade-off between accuracy and efficiency and has better deployment support.

3.Evaluate the model for language-heavy or language-only task, where the visual component plays only an auxiliary role or is entirely absent.
Report memory consumption for training.

---

> ### Author Response · Authors · 2025-11-21
> **To Reviewer x6UG (Weakness)**
>
> > (W1) The performance gains at higher bit-widths (e.g., INT3) are relatively small — sometimes less than 0.2%, and even slightly inferior to baselines on specific models such as Qwen2-VL-2B-INT3g128.
>
> We thank the reviewer for the thoughtful observation and apologize for not discussing this phenomenon more clearly in the original manuscript. When quantizing to relatively higher bit-widths such as INT3g128, the quantized model already operates very close to its FP16 counterpart, leaving limited room for further improvement. In such near–floating-point regimes, quantization noise is inherently small, the robust model capacity is largely preserved, and the upper-bound performance is dictated by the full-precision model itself. As a result, our method yields only marginal gains. We would like to emphasize that these minor performance differences at higher bit-widths do not diminish the validity of our approach, even in ultra–low-bit scenarios such as INT2g128. Rather, they reflect the inherent challenge of obtaining further improvements when the model is already operating close to full-precision accuracy. If additional gains at higher bit-widths are desired, integrating VLMQ with other complementary techniques may offer potential benefits, and we consider this an interesting direction for future exploration. In addition, we provide the INT3 results in Appendix Table 8 of the original paper. We also note that the improvements offered by VLMQ would likely be more pronounced under INT3 if groupwise quantization were not applied.
>
> We have added clarifications in the revised manuscript and sincerely apologize for any confusion caused.
>
> > (W2) The method introduces additional computational and memory overhead due to the need for block-wise backpropagation, which limits its efficiency compared to simpler post-training quantization approaches like GPTQ.
>
> We thank the reviewer for the comment. Please refer to our response to W3 for additional details.
>
> > (W3) It is not clear how the method performs for language-heavy/language-only task. Given that GPTQ is relatively outdated for language tasks, it would be helpful to examine how the proposed token-importance weighting impacts language performance.
>
> We thank the reviewer for pointing this out. We have added the relevant experiments, and the corresponding details can be found in our response to Q3.

---

> ### Author Response · Authors · 2025-11-21
> **To Reviewer x6UG (Questions)**
>
> > (Q1) Include experiments on larger models such as LLaVA-13B or InternVL-26B to verify scalability.
>
> We thank the reviewer for the suggestion. To assess scalability on larger models, we report the INT3 quantization performance of Qwen2.5-VL-**32B**-Instruct, as shown below.
>
> | Method | Avg ($\uparrow$)|
> |:------:|:---------------:|
> | Full Prec | 75.51 |
> | GPTQ | 74.20 |
> | GPTAQ | 73.68 |
> | VLMQ | **74.31** |
>
> The fact that VLMQ narrows the gap to full precision while maintaining stability suggests that the proposed method scales well with model size. The full results have been added to the revised manuscript.
>
> > (Q2) Provide results for INT4 quantization, which may offer a better trade-off between accuracy and efficiency and has better deployment support.
>
> We thank the reviewer for the suggestion. Please find the INT4 quantization results for Qwen2-VL-7B-Instruct below.
>
> *Average accuracy on Qwen2-VL-7B-Instruct-INT4*
> | Method | Avg ($\uparrow$)|
> |:------:|:---------------:|
> | Full Prec | 75.81 |
> | GPTQ | 73.25 |
> | GPTAQ | 73.63 |
> | VLMQ | **73.87 $_\text{+0.24\%}$** |
>
> We have added these INT4 results to the revised manuscript, and additional INT4 experiments on other models are currently in progress.
>
> > (Q3) Evaluate the model for language-heavy or language-only task, where the visual component plays only an auxiliary role or is entirely absent. Report memory consumption for training.
>
> We thank the reviewer for the suggestion. To assess the behavior of VLMQ on language-heavy and language-only tasks, we evaluate Qwen2-VL-7B-Instruct-INT2g128 on HellaSwag [4] (language-only) and TextCaps [3] (language-heavy). The results are presented below.
>
> |Method|HellaSwag (Accuracy (%))|TextCaps (Val) (CIDEr $\times$ 100)|
> |:---:|:---:|:---:|
> |Full Prec|77.63|152.00|
> |GPTQ|51.29|55.06|
> |GPTAQ|49.82|46.52|
> |VLMQ|53.19|55.90|
>
> Across both benchmarks, the quantized model retains strong performance, indicating that VLMQ does not compromise the model’s core language understanding or generation abilities. These findings have been incorporated into the revised version of the manuscript.
>
> **Memory consumption for training.** We would like to emphasize that VLMQ does not introduce any training or fine-tuning. Our method is a pure post-training quantization procedure, and the block-wise backward operation is used solely for gradient extraction. As correctly pointed out in W2, our method performs a one-pass block-wise backpropagation step to extract gradient information, which does introduce additional computational and memory overhead compared to purely post-training quantization methods such as GPTQ. However, we emphasize that this backpropagation is lightweight and remains highly efficient in practice. We are able to quantize a 7B model with negligible additional latency, and a 72B model can still be processed on a single 80GB H200 GPU, demonstrating both computational and memory efficiency. The corresponding evaluation results are provided in the following:
>
> | Model Size | Metrice | GPTQ | GPTAQ | VLMQ |
> |:----------:|:-------:|:----:|:-----:|:----:|
> |2B|Quant time (h)|0.05|0.07|0.09 $_{\text{+1.2 mins}}$|
> |2B|Peak memory (GB)|4.99|10.59|12.78 $_{\text{+2.19 GB}}$|
> |7B|Quant time (h)|0.08|0.18|0.21 $_{\text{+1.8 mins}}$|
> |7B|Peak memory (GB)|17.37|24.76|29.05 $_{\text{+4.29 GB}}$|
> |32B|Quant time (h)|0.31|0.84|0.94 $_{\text{+6.0 mins}}$|
> |32B|Peak memory (GB)|18.99|38.94|41.56 $_{\text{+2.62 GB}}$|
>
> All quantization experiments were run on a single 80GB NVIDIA H100, although most settings are also feasible on GPUs with considerably less memory. Importantly, the additional quantization time and incremental memory footprint introduced by VLMQ are both relatively small, indicating that our method does not incur substantial overhead in practice.
>
> [1] Li, Shiyao, et al. "MBQ: Modality-balanced quantization for large vision-language models." Proceedings of the Computer Vision and Pattern Recognition Conference. 2025.
>
> [2] Zheng, Xingyu, et al. "First-Order Error Matters: Accurate Compensation for Quantized Large Language Models." arXiv preprint arXiv:2507.11017 (2025).
>
> [3] Sidorov, Oleksii, et al. "Textcaps: a dataset for image captioning with reading comprehension." European conference on computer vision. Cham: Springer International Publishing, 2020.
>
> [4] Zellers, Rowan, et al. "Hellaswag: Can a machine really finish your sentence?." arXiv preprint arXiv:1905.07830 (2019).

---

### Official Review · Reviewer_ZiWz · 2025-11-01

**Soundness:** 3
**Presentation:** 3
**Contribution:** 2
**Rating:** 4
**Confidence:** 5

**Summary:**

This paper proposes a post-training quantization (PTQ) framework tailored for visual-language models (VLMs) based on existing PTQ techniques. It first identifies two critical issues in quantizing VLM models: visual over-representation and modality gap. Building upon these challenges, the paper introduces a diagonal gradient-derived importance factor to reflect token-level significance and employs a single lightweight block-wise backpropagation to compute gradients. The paper's observations on the characteristics of VLM model activation inputs offer valuable insights, and the approach of computing token importance from gradients warrants further investigation.

**Strengths:**

The paper clearly identifies two overlooked challenges in post-training quantization (PTQ) for vision-language models (VLMs): visual over-representation and modality gap. These insights are well-motivated and supported by empirical visualization. The proposed VLMQ framework introduces a gradient-driven importance factor that selectively emphasizes salient tokens and suppresses redundant ones. This token-wise quantization adjustment is both theoretically justified (Theorem 4.1) and efficiently implemented via block-wise backpropagation, striking a good balance between accuracy and computational cost.

**Weaknesses:**

1. Limited Analysis of Generalization and Robustness
While results cover multiple benchmarks, the study focuses on VQA-style tasks. It remains unclear whether the proposed method generalizes equally well to other multimodal reasoning or generation tasks (e.g., captioning, retrieval, or multi-image reasoning). Moreover, the paper’s gradient-based importance analysis is overly simplistic and lacks comparison with the Taylor-expansion-based importance estimation employed in SliM-LLM[1]. Additionally, the description of the block-wise backpropagation mechanism is insufficiently detailed. A thorough comparison and analysis against existing block-wise quantization methods (e.g., Q-VLM) would significantly strengthen this section.

2. Computational Efficiency Claims Need More Clarity
The paper claims that block-wise backpropagation is “lightweight”, but only provides GPU-hour comparisons in small-scale settings. A detailed runtime and memory analysis on full-scale 32B models would better substantiate the claimed efficiency. Furthermore, the paper omits any evaluation of the memory footprint and inference speedup achieved after quantization. Such empirical results are essential and should be included to fully support the efficiency claims.

3. Lack of Discussion on Integration with Other Quantization Frameworks
Although the authors mention that VLMQ can be combined with other PTQ algorithms, experiments only integrate it with GPTQ/GPTAQ. Broader evaluations (e.g., with SmoothQuant or Q-VLM) would strengthen the argument for framework compatibility, and better demonstrate the superiority of the VLMQ approach in terms of performance and computational overhead.

[1]Huang, Wei, et al. "SliM-LLM: Salience-driven mixed-precision quantization for large language models." arXiv preprint arXiv:2405.14917 (2024).

**Questions:**

Further question:
1.	Can token-wise importance estimation be extended to dynamic or streaming multimodal inputs? Since current VLMQ computes importance factors from static calibration data, investigating adaptive or online estimation methods could improve robustness when processing diverse, real-world multimodal streams. Is it feasible to compute token importance in real time based on gradients during dynamic input processing?
2.	Can importance-aware quantization be integrated with training-time or mixed-precision quantization strategies? Exploring how the proposed token-level weighting interacts with quantization-aware training (QAT) or hybrid bit-width assignments might further improve model efficiency without sacrificing accuracy.
3.	What are the effects of token-wise quantization on cross-modal alignment and interpretability? Since VLMQ adjusts the relative weighting of tokens, analyzing how it alters attention distributions or latent alignment between text and vision could offer deeper insights into multimodal representation learning. Addressing this question not only strengthens the theoretical foundation of this paper but also facilitates further research on MLLMs.

---

> ### Author Response · Authors · 2025-11-20
> **Response to Reviewer ZiWz (Part 1)**
>
> > Limited Analysis of Generalization and Robustness While results cover multiple benchmarks, the study focuses on VQA-style tasks. It remains unclear whether the proposed method generalizes equally well to other multimodal reasoning or generation tasks (e.g., captioning, retrieval, or multi-image reasoning).
>
> To better understand the generalization ability of VLMQ, we additionally evaluate MMMU (Val) and TextCaps (Val) on Qwen2-VL-7B-Instruct-INT2g128. The results are shown below.
>
> |Method|MMMU (Val) (Accuracy (%))|TextCaps (Val) (CIDEr $\times$ 100)|
> |:---:|:---:|:---:|
> |Full Prec|50.44|152.00|
> |GPTQ|32.22|55.06|
> |GPTAQ|22.25|46.52|
> |VLMQ|**33.67**|**55.90**|
>
> These results demonstrate that VLMQ generalizes effectively beyond VQA-style tasks. These findings show that VLMQ maintains strong performance across heterogeneous multimodal tasks, supporting its ability to generalize well beyond the specific settings evaluated in the main experiments.
>
> > Moreover, the paper’s gradient-based importance analysis is overly simplistic and lacks comparison with the Taylor-expansion-based importance estimation employed in SliM-LLM [1].
>
> We thank the reviewer for pointing this out. We would like to gently clarify that the motivation and design principles of VLMQ are substantially different from those of SliM-LLM, and therefore a direct comparison may not fully reflect the intended purposes of the two methods.
>
> - Design scope. SliM-LLM focuses on mixed-precision quantization and aims to allocate bitwidths across weight groups, whereas VLMQ is designed for single-precision quantization (e.g., INT2/INT3) and targets robustness under extremely low-bit settings. Because the two methods were developed with different goals in mind, a direct comparison may not provide a fair or complete picture.
>
> - Use of gradients. Although both SliM-LLM and VLMQ leverage gradient information, the underlying motivations are entirely different. SliM-LLM inherits the importance metric from SparseGPT [2], using second-order Taylor approximations to identify salient weight elements for mixed-precision bitwidth allocation. In contrast, VLMQ applies first-order gradient information that is supported by the Taylor expansion in Theorem 4.1 to identify redundant or low importance vision tokens (i.e., activations). This aligns with our motivation of mitigating visual over-representation and addressing the modality-gap issue in multimodal LLMs. Our goal is not to allocate bitwidths to weights, but to adjust token contributions so that the calibration data distribution becomes more representative for quantization. The theoretical justification (Theorem 4.1) together with experimental evidence (Figure 3) demonstrates that first-order gradients serve as an effective indicator for identifying redundant tokens.
>
> Overall, the two approaches address fundamentally different problems: weight salience for mixed-precision allocation versus activation redundancy for calibration correction, which naturally leads to distinct uses of gradient information. We will present these distinctions more clearly in the revised manuscript, along with the corresponding analysis.

---

> ### Author Response · Authors · 2025-11-20
> **Response to Reviewer ZiWz (Part 2)**
>
> > Additionally, the description of the block-wise backpropagation mechanism is insufficiently detailed. A thorough comparison and analysis against existing block-wise quantization methods (e.g., Q-VLM) would significantly strengthen this section.
>
> We apologize for any confusion caused.
>
> **Block-wise backpropagation mechanism.** Below we provide a more detailed explanation of the block-wise backpropagation procedure used in VLMQ. Taking a standard attention block as an example, given the activation $\hat{X} \in \mathbb{R}^{N \times C_i}$, the attention computation is formulated as
>
> $$\hat{O} = \hat{X} + \texttt{Attn}(\hat{X}),$$
>
> where $\texttt{Attn}(\cdot)$ denotes the standard causal attention employed in LLMs. After obtaining the block output $\hat{O}$, we compute the block-wise distillation loss using mean-squared error:
>
> $$L_\text{MSE} = \text{MSE}(O, \hat{O}),$$
>
> where $O$ is the full-precision supervision. We then perform a one-pass backpropagation within the block only to obtain gradients of the post-projection activations $Q, K, V, O \in \mathcal{R}^{N, C_o}$. Their corresponding raw gradients $P_Q, P_K, P_V, P_O$ are cached via backward hooks, with the same shape as their activations. These gradients are subsequently processed according to Eq. 4 in the manuscript. Importantly, this backward pass is used solely for gradient extraction—no parameter updates are performed.
>
> **Clarification regarding "block-wise" methods.** We would also like to clarify that the notion of “block-wise” in VLMQ is conceptually different from that used in methods such as Q-VLM. In VLMQ, block-wise refers specifically to using architectural blocks as the minimal units for gradient extraction in order to estimate token-level importance. This is fundamentally distinct from post-quantization block-wise reconstruction techniques commonly used in CNN/ViT quantization [1, 3–7] or in certain LLM quantization approaches [8, 9], which refine weight or scale parameters within predefined blocks and are typically much more computationally demanding. Q-VLM belongs to this latter category, where the focus is on optimizing block partitioning strategies for reconstruction rather than estimating token importance. Our method therefore serves a different purpose and operates under a different paradigm.
>
> In summary, the “block-wise” mechanism in VLMQ is a lightweight gradient-caching strategy designed solely to facilitate token-level importance estimation, whereas Q-VLM and related reconstruction-based approaches target parameter adjustment within quantization blocks.

---

> ### Author Response · Authors · 2025-11-20
> **Response to Reviewer ZiWz (Part 3)**
>
> > Computational Efficiency Claims Need More Clarity The paper claims that block-wise backpropagation is “lightweight”, but only provides GPU-hour comparisons in small-scale settings. A detailed runtime and memory analysis on full-scale 32B models would better substantiate the claimed efficiency.
>
> Thanks for pointing this out. We present the peak memory and quantization time for 2B/7B/32B models respectively.
>
> | Model Size | Metrice | GPTQ | GPTAQ | VLMQ |
> |:----------:|:-------:|:----:|:-----:|:----:|
> |2B|Quant time (h)|0.05|0.07|0.09 $_{\text{+1.2 mins}}$|
> |2B|Peak memory (GB)|4.99|10.59|12.78 $_{\text{+2.19 GB}}$|
> |7B|Quant time (h)|0.08|0.18|0.21 $_{\text{+1.8 mins}}$|
> |7B|Peak memory (GB)|17.37|24.76|29.05 $_{\text{+4.29 GB}}$|
> |32B|Quant time (h)|0.31|0.84|0.94 $_{\text{+6.0 mins}}$|
> |32B|Peak memory (GB)|18.99|38.94|41.56 $_{\text{+2.62 GB}}$|
>
> All quantization experiments were run on a single 80GB NVIDIA H100, although most settings are also feasible on GPUs with considerably less memory. Importantly, the additional quantization time and incremental memory footprint introduced by VLMQ are both relatively small, indicating that our method does not incur substantial overhead in practice.
>
> > Furthermore, the paper omits any evaluation of the memory footprint and inference speedup achieved after quantization. Such empirical results are essential and should be included to fully support the efficiency claims.
>
> Thank you for raising this point. We would like to clarify that our method introduces optimizations only during the quantization process and does not modify the underlying quantization format used by GPTQ. In other words, VLMQ preserves the original `quant()` representation and weight layout, ensuring full compatibility with existing GPTQ-style deployment pipelines. Consequently, models quantized with VLMQ can directly leverage current hardware-optimized kernels—such as the GPTQ-released W3 kernel, Marlin, and ExLLaMA—without requiring any changes to inference backends or incurring additional runtime overhead.
>
> In addition to maintaining full format-level compatibility, we are also working toward supporting deployment within widely used acceleration frameworks such as AutoGPTQ. To provide a basic demonstration of inference efficiency, we benchmarked the official GPTQ W3 kernel optimized on an RTX 5090 GPU and evaluated several major linear layers of Qwen2-VL-7B (INT3), including `qkv_proj`, `out_proj`, `gateup_proj`, and `down_proj`. The results are shown below:
>
> Linear Type| Size of $W$ (in_dim $\times$ out_dim) | FP16 ($\mu s$) | W3 ($\mu s$) | Speedup |
> |:---:|:---:|:---:|:---:|:---:|
> `qkv_proj`|(3584, 4608)|38.97|4.17|9.34 $\times$|
> `out_proj`|(3584, 3584)|39.00|4.17|9.34 $\times$|
> `gateup_proj`|(3584, 37888)|175.17|18.46|9.49 $\times$|
> `down_proj`|(18944, 3584)|96.39|10.47|9.20 $\times$|
>
> These measurements show that 3-bit quantization using the standard GPTQ format retained in VLMQ already delivers substantial acceleration, approximately ~9 $\times$ speedup on representative linear layers without requiring any modifications to existing hardware kernels. This confirms that VLMQ retains GPTQ’s deployment efficiency and can seamlessly benefit from optimized low-bit inference pipelines.

---

> ### Author Response · Authors · 2025-11-20
> **Response to Reviewer ZiWz (Part 4)**
>
> > Lack of Discussion on Integration with Other Quantization Frameworks Although the authors mention that VLMQ can be combined with other PTQ algorithms, experiments only integrate it with GPTQ/GPTAQ. Broader evaluations (e.g., with SmoothQuant or Q-VLM) would strengthen the argument for framework compatibility, and better demonstrate the superiority of the VLMQ approach in terms of performance and computational overhead.
>
> We thank the reviewer for raising this point. Our work focuses on developing an efficient weight-only quantization method tailored for large VLMs. Methods such as SmoothQuant and Q-VLM fall under the category of weight–activation quantization, which involves fundamentally different design considerations and is therefore beyond the scope of this paper. Extending VLMQ to weight–activation quantization would require addressing activation outliers and a different calibration pipeline, which we regard as an interesting direction for future work. For other weight-only approaches such as AWQ, we observed experimentally that AWQ collapses under the INT2g128 setting, and MBQ—built upon AWQ—exhibits the same failure pattern. In contrast, the GPTQ series maintains the basic model capabilities under this configuration. Similar instability has also been independently reported in ApiQ (Table 3) [10] and SliM-LLM (Table 1) [11]. These observations underscore an important point: the quality and stability of the precursor PTQ algorithm significantly influence the final performance after integration with VLMQ. To illustrate this, we provide the following comparison:
>
> *Average accuracy on Qwen2.5-VL-7B-Instruct-INT2g128*
> | Method | Avg ($\uparrow$)|
> |:------:|:---------------:|
> | GPTQ | 55.58 |
> | GPTQ+VLMQ | **57.46** |
> | GPTAQ | 41.69 |
> | GPTAQ+VLMQ | **52.10** |
>
> While VLMQ improves GPTAQ by roughly 10 points, the GPTAQ baseline itself is significantly degraded under INT2g128, making it unsuitable as a strong foundation. For this reason, we adopt GPTQ as the primary precursor for integrating VLMQ. Given the instability or incompatibility of other approaches under INT2 settings, we believe that further attempts beyond state-of-the-art weight-only PTQ methods are not necessary within the scope of this work.

---

> ### Author Response · Authors · 2025-11-20
> **Response to Reviewer ZiWz (Part 5)**
>
> > Can token-wise importance estimation be extended to dynamic or streaming multimodal inputs? Since current VLMQ computes importance factors from static calibration data, investigating adaptive or online estimation methods could improve robustness when processing diverse, real-world multimodal streams. Is it feasible to compute token importance in real time based on gradients during dynamic input processing?
>
> We thank the reviewer for raising this interesting question. We are happy to discuss this point, and it is indeed an active direction we are currently exploring. In short, the VLMQ-style gradient-based importance estimation is not directly suitable for real-time or streaming applications (e.g., streaming QA), as it requires online backpropagation, which would be computationally prohibitive in such scenarios.
>
> Fortunately, the community has already proposed several promising approaches for real-time token importance estimation without requiring gradients. For instance, FastV [12] leverages attention scores to identify salient tokens, while DivPrune [13] exploits token-level diversity to filter redundant information. These methods indicate that token salience can be estimated effectively without backpropagation. As a takeaway, we also explored whether these inference-time importance estimation metrics could be applied to improve VLMQ. Unfortunately, although they are effective for online inference, our ablation results (Table 4) show that they do not fully capture the underlying token importance required for quantization calibration. In contrast, our gradient-based estimation is theoretically grounded (Theorem 4.1), empirically validated (Figure 3), and consistently leads to better quantization performance.
>
> It is also worth noting that most existing online vision–language QA tasks still follow a "one-pass inference" paradigm in which the entire video and question are available beforehand. In such settings, it is possible to design query-conditioned importance estimation or activation compression strategies. However, the situation changes substantially in streaming scenarios.
>
> Streaming multimodal token processing differs from offline QA in two fundamental aspects:
> - the query is not present during the estimation phase, and
> - the interaction is multi-turn, requiring the model to preserve and manage evolving context.
> To support these scenarios, importance estimation must be query-agnostic, lightweight, and capable of operating on-the-fly.
>
> Recent works such as ReKV [15], StreamKV [14], StreamMem [16], LiveVLM [17], and InfiniPot-V [18] have explored query-free token importance assignment or long-context compression for streaming inference. For example, ReKV stores all key-value (KV) pairs by offloading them to CPU memory and retrieves necessary context when needed. StreamKV and StreamMem attempt to address the absence of queries by introducing general-purpose proxy tokens or prompts to estimate token importance online and enable further compression.
>
> Overall, achieving streaming token-wise importance estimation would require query-agnostic, real-time token salience estimation techniques that balance activation distortion with inference efficiency. We view this as a valuable and exciting direction for future research.

---

> ### Author Response · Authors · 2025-11-20
> **Response to Reviewer ZiWz (Part 6)**
>
> > Can importance-aware quantization be integrated with training-time or mixed-precision quantization strategies? Exploring how the proposed token-level weighting interacts with quantization-aware training (QAT) or hybrid bit-width assignments might further improve model efficiency without sacrificing accuracy.
>
> We thank the reviewer for these insightful questions.
>
> **Compatibility with mixed-precision quantization.** The proposed importance-aware mechanism can, in principle, be integrated with mixed-precision quantization frameworks and follow-up work in this direction is currently in progress.
>
> **Compatibility with QAT.** From an engineering perspective, integrating our approach with QAT is certainly feasible. However, we note that QAT already incorporates token-level importance implicitly through supervised fine-tuning or distillation losses. During QAT, gradients are computed with respect to the task loss, which naturally assigns higher weight to tokens that contribute more strongly to task performance. In addition, teacher–student distillation further amplifies this effect by encouraging the model to match the teacher’s attention patterns and token sensitivities, thereby embedding token importance into the training dynamics without requiring an explicit weighting mechanism. The actual benefit of explicitly adding importance-aware weighting on top of QAT would therefore require additional investigation. Given our limited computational resources, we are unable to conduct a full QAT study within the scope of this submission, but we agree that this is an interesting direction for future exploration.
>
> > What are the effects of token-wise quantization on cross-modal alignment and interpretability? Since VLMQ adjusts the relative weighting of tokens, analyzing how it alters attention distributions or latent alignment between text and vision could offer deeper insights into multimodal representation learning. Addressing this question not only strengthens the theoretical foundation of this paper but also facilitates further research on MLLMs.
>
> We appreciate the reviewer’s insightful question regarding the potential impact of token-wise importance-aware quantization on cross-modal alignment and interpretability. Conceptually, VLMQ operates by down-weighting redundant vision tokens while preserving the contribution of salient, semantically meaningful tokens. By reducing visual over-representation and suppressing background noise, VLMQ is designed to reinforce rather than disrupt the alignment between visual and textual modalities. Empirically, our results show consistent improvements on multiple cross-modal benchmarks that rely heavily on fine-grained vision–language alignment. These performance gains indicate that the core cross-modal interactions remain intact, and in many cases are even strengthened, after applying importance-aware quantization. The behavior is consistent with our motivation: emphasizing essential visual cues helps the model focus on the most relevant information, thereby supporting both alignment quality and interpretability.

---

> ### Author Response · Authors · 2025-11-20
> **Response to Reviewer ZiWz (References)**
>
> ### Reference
>
> [1] Li, Yuhang, et al. "Brecq: Pushing the limit of post-training quantization by block reconstruction." arXiv preprint arXiv:2102.05426 (2021).
>
> [2] Frantar, Elias, and Dan Alistarh. "Sparsegpt: Massive language models can be accurately pruned in one-shot." International conference on machine learning. PMLR, 2023.
>
> [3] Wei, Xiuying, et al. "Qdrop: Randomly dropping quantization for extremely low-bit post-training quantization." arXiv preprint arXiv:2203.05740 (2022).
>
> [4] Lv, Chengtao, et al. "Ptq4sam: Post-training quantization for segment anything." Proceedings of the IEEE/CVF Conference on computer vision and pattern recognition. 2024.
>
> [5] Li, Yuhang, and Priyadarshini Panda. "Tesseraq: Ultra low-bit llm post-training quantization with block reconstruction." arXiv preprint arXiv:2410.19103 (2024).
>
> [6] Wu, Zhuguanyu, et al. "APHQ-ViT: Post-Training Quantization with Average Perturbation Hessian Based Reconstruction for Vision Transformers." Proceedings of the Computer Vision and Pattern Recognition Conference. 2025.
>
> [7] Wu, Zhuguanyu, et al. "FIMA-Q: Post-Training Quantization for Vision Transformers by Fisher Information Matrix Approximation." Proceedings of the Computer Vision and Pattern Recognition Conference. 2025.
>
> [8] Ding, Xin, et al. "Cbq: Cross-block quantization for large language models." arXiv preprint arXiv:2312.07950 (2023).
>
> [9] Shao, Wenqi, et al. "Omniquant: Omnidirectionally calibrated quantization for large language models." arXiv preprint arXiv:2308.13137 (2023).
>
> [10] Liao, Baohao, et al. "ApiQ: Finetuning of 2-bit quantized large language model." arXiv preprint arXiv:2402.05147 (2024).
>
> [11] Huang, Wei, et al. "SliM-LLM: Salience-driven mixed-precision quantization for large language models." arXiv preprint arXiv:2405.14917 (2024).
>
> [12] Chen, Liang, et al. "An image is worth 1/2 tokens after layer 2: Plug-and-play inference acceleration for large vision-language models." European Conference on Computer Vision. Cham: Springer Nature Switzerland, 2024.
>
> [13] Alvar, Saeed Ranjbar, et al. "Divprune: Diversity-based visual token pruning for large multimodal models." Proceedings of the Computer Vision and Pattern Recognition Conference. 2025.
>
> [14] Chen, Yilong, et al. "StreamKV: Streaming Video Question-Answering with Segment-based KV Cache Retrieval and Compression." arXiv preprint arXiv:2511.07278 (2025).
>
> [15] Di, Shangzhe, et al. "Streaming video question-answering with in-context video kv-cache retrieval." arXiv preprint arXiv:2503.00540 (2025).
>
> [16] Yang, Yanlai, et al. "Streammem: Query-agnostic kv cache memory for streaming video understanding." arXiv preprint arXiv:2508.15717 (2025).
>
> [17] Ning, Zhenyu, et al. "LiveVLM: Efficient Online Video Understanding via Streaming-Oriented KV Cache and Retrieval." arXiv preprint arXiv:2505.15269 (2025).
>
> [18] Kim, Minsoo, et al. "InfiniPot-V: Memory-Constrained KV Cache Compression for Streaming Video Understanding." arXiv preprint arXiv:2506.15745 (2025).

---

### Official Review · Reviewer_RFpm · 2025-11-01

**Soundness:** 3
**Presentation:** 3
**Contribution:** 2
**Rating:** 4
**Confidence:** 4

**Summary:**

This paper focuses on post-training quantization (PTQ) for Vision-Language Models (VLMs). One key observation is that VLM activations exhibit two intrinsic characteristics: visual over-representation and modality gap, which have been overlooked by existing PTQ methods. A VLM-tailored PTQ framework, VLMQ, is proposed to mitigate these issues. Experiments are carried out on 8 benchmarks across VLMs of various sizes.

**Strengths:**

[+] The manuscript is well written, with clear logics.

[+] The symbol definitions are clear, and the image visualization is complete.

[+] Many experiments are conducted to analyze the proposed method.

**Weaknesses:**

[-] The core ablation experiments are insufficient. The key contribution is introducing a weighted strategy based on the importance score, thus one critical ablation experiment should be: baseline (e.g., GPTQ/GPTAQ) vs. VLMQ vs. VLMQ but removing importance weighting (i.e., all tokens have equal weights). This constitutes a logical gap in the argumentation.

[-] Under INT2 settings, the results of AWQ and MBQ methods almost completely collapse (e.g. accuracy of 0.00% or single digits). Although this highlights the superiority of VLMQ, it also raises the question: are these two methods really that fragile? Or did the author not use their officially recommended parameters or implementations optimized for ultra-low bit rates?

[-] The fairness and consistency of baseline comparisons. Across different experimental setups, the paper switches the baselines. Why was a weaker method, GPTQ, chosen as the baseline for the INT2 setting where performance is most critical, rather than the stronger GPTAQ? This raises doubts that the VLMQ approach may be incompatible with certain mechanisms of GPTAQ or perform poorly when combined. A fairer comparison would involve applying VLMQ to both GPTQ and GPTAQ separately and then comparing the results with the original GPTQ and GPTAQ.

**Questions:**

[-] INT3g128: On Qwen2-VL-2B, the average score of VLMQ (62.90) is lower than that of GPTAQ (63.44). The author attributes it to the abnormal performance of GPTAQ on MME RealWorld, but this does not change the fact that VLMQ is not optimal in this setting. INT2g128: On the LLaVA-OneVision-7B model, VLMQ's performance on MMEen (38.66) was significantly lower than GPTQ (40.29) and GPTAQ (42.59), but it surpassed the baseline by a slight advantage on other tasks in the final average score. This situation of "overall victory but lagging behind in key indicators" should be discussed in more depth.

---

> ### Author Response · Authors · 2025-11-20
>
> > The core ablation experiments are insufficient. The key contribution is introducing a weighted strategy based on the importance score, thus one critical ablation experiment should be: baseline (e.g., GPTQ/GPTAQ) vs. VLMQ vs. VLMQ but removing importance weighting (i.e., all tokens have equal weights). This constitutes a logical gap in the argumentation.
>
> We thank the reviewer for the valuable suggestion. Following the reviewer’s comment, we conduct a dedicated study to directly evaluate the contribution of the proposed importance-weighted strategy. Specifically, we compare four variants on Qwen2-VL-Instruct-INT2g128: GPTQ, GPTAQ, our full VLMQ, and VLMQ without importance weighting (denoted as VLMQ $_{\text{naive}}$).
>
> In VLMQ $_{\text{naive}}$, a subset of vision tokens is randomly assigned as low-importance (LI), and a uniform down-weighting factor (DW Factor) is applied, following the reviewer’s suggestion that all tokens should be treated equally in the ablation. We perform a comprehensive 3 × 3 grid search over LI Ratio and DW Factor. The accuracies on TextVQA and DocVQA as supplementary experiments of preliminary studies (Table 1) are summarized below:
>
> | Mehod | LI Ratio | DW Factor | TextVQA | DocVQA |
> |:-:|:-:|:-:|:-:|:-:|
> |Full Prec|-|-|82.02|93.89|
> |GPTQ|-|-|71.93|74.90|
> |GPTAQ|-|-|67.98|72.57|
> |VLMQ $_{\text{naive}}$|0.25|0.01|70.66|72.12|
> |VLMQ $_{\text{naive}}$|0.25|0.05|72.12|72.63|
> |VLMQ $_{\text{naive}}$|0.25|0.10|70.89|72.10|
> |VLMQ $_{\text{naive}}$|0.50|0.01|72.38|72.76|
> |VLMQ $_{\text{naive}}$|0.50|0.05|69.83|71.93|
> |VLMQ $_{\text{naive}}$|0.50|0.10|71.79|73.67|
> |VLMQ $_{\text{naive}}$|0.75|0.01|69.74|69.44|
> |VLMQ $_{\text{naive}}$|0.75|0.05|68.17|71.05|
> |VLMQ $_{\text{naive}}$|0.75|0.10|71.21|70.33|
> |**VLMQ (Ours)**|**Grad-driven**|**Grad-driven**|**74.80**|**75.76**|
>
> These results reveal two important findings:
> 1.	Even coarse, randomly assigned token weighting (VLMQ $_{\text{naive}}$) improves over GPTQ/GPTAQ in most cases, indicating that the general idea of selectively down-weighting redundant vision tokens is effective.
> 2.	Our full gradient-driven VLMQ achieves the best performance, demonstrating that a finer-grained, importance-aware strategy is essential for fully exploiting vision token redundancy.
>
> We have added this ablation and its analysis to the revised manuscript to address the reviewer’s concern and close the logical gap.
>
> > Under INT2 settings, the results of AWQ and MBQ methods almost completely collapse (e.g. accuracy of 0.00% or single digits). Although this highlights the superiority of VLMQ, it also raises the question: are these two methods really that fragile? Or did the author not use their officially recommended parameters or implementations optimized for ultra-low bit rates?
>
> We thank the reviewer for raising this important point. To ensure a fair comparison, we reproduced MBQ using its official repository and evaluated AWQ using the integrated codebase, which directly inherits from the official AWQ implementation. All experiments strictly follow the recommended parameter settings provided in the original papers. It is worth noting that neither MBQ [1] nor AWQ [2] reports INT2-g128 results, making our reproduction essential for establishing their behavior under ultra–low-bit quantization.
>
> To further verify whether these methods are indeed fragile under ultra–low-bit settings, we refer to ApiQ [3], where the authors report Perplexity (PPL) results on WikiText and C4 for INT2g128 quantization on LLaMA-2-7B and LLaMA-2-13B (Table 3 in [3]). We list those reported numbers below:
>
> | Method | 7B PPL@Wiki | 7B PPL@C4 | 13B PPL@Wiki | 13B PPL@C4 |
> |:-:|:-:|:-:|:-:|:-:|
> |GPTQ|36.77|33.70|28.14|20.97|
> |AWQ|2.2e5|1.7e5|1.2e5|9.4e4|
>
> A similar trend is observed in SliM-LLM [4], where the authors report WikiText PPL for the LLaMA-1, LLaMA-2, and LLaMA-3 INT2g128 series. The corresponding results are summarized below:
>
> | Method | 1-7B | 1-13B | 1-30B | 1-65B | 2-7B | 2-13B | 2-70B | 3-8B | 3-70B |
> |:-:|:-:|:-:|:-:|:-:|:-:|:-:|:-:|:-:|:-:|
> |GPTQ|152.31|20.44|13.01|9.51|60.45|28.14|8.78|210.00|11.90|
> |AWQ|2.6e5|2.8e5|2.4e5|7.4e4|2.2e5|1.2e5|-|1.7e6|1.7e6|
>
>
> These findings reveal a consistent pattern: 1) GPTQ maintains reasonable performance even at 2-bit granularity, whereas 2) AWQ collapses dramatically, with PPL increasing by orders of magnitude. This strongly suggests that activation-aware methods such as AWQ and methods derived from it are inherently unstable under ultra-low-bit quantization. MBQ, which inherits AWQ’s activation-weighting mechanism, naturally suffers from similar catastrophic degradation.
>
> Therefore, our observed results (near-zero accuracy for AWQ/MBQ under INT2 settings) are consistent with independent evidence from existing findings, rather than stemming from improper parameter choices or incorrect implementation.

---

> > ### Author Response · Authors · 2025-11-20
> >
> > > The fairness and consistency of baseline comparisons. Across different experimental setups, the paper switches the baselines. Why was a weaker method, GPTQ, chosen as the baseline for the INT2 setting where performance is most critical, rather than the stronger GPTAQ? This raises doubts that the VLMQ approach may be incompatible with certain mechanisms of GPTAQ or perform poorly when combined. A fairer comparison would involve applying VLMQ to both GPTQ and GPTAQ separately and then comparing the results with the original GPTQ and GPTAQ.
> >
> > We thank the reviewer for the insightful comments. The key distinction between GPTQ and GPTAQ lies in their calibration pipelines. GPTQ adopts a symmetric, layer-isolated calibration, whereas GPTAQ introduces an asymmetric, sequential calibration that explicitly incorporates accumulated quantization errors from preceding layers.
> >
> > More concretely, for a model with $L$ linear layers:
> > - GPTQ calibrates each layer independently. When quantizing the $l$-th layer, it always uses full-precision inputs $X$ as calibration data, without accounting for quantization noise propagated from the previous $l-1$ layers.
> > - GPTAQ, in contrast, calibrates in a strictly sequential layer-wise manner.
> > The $l$-th layer is calibrated using noisy, quantized inputs $\hat{X}$ that already contain accumulated errors $E$ introduced by the previously quantized layers.
> >
> > This sequential mechanism enables GPTAQ to outperform GPTQ in moderate quantization regimes such as INT3, where the propagated error remains within a tolerable range and can help the model adapt to quantization noise. However, when the bit-width is further reduced to INT2, the accumulated error becomes substantially larger. In this extreme regime, the calibration input $\hat{X}$ fed into GPTAQ is heavily biased due to upstream distortion. As a result, the sequential optimization causes the weight distribution of the current layer to be calibrated based on severely corrupted inputs, ultimately harming performance. GPTQ, which does not rely on noisy inputs, is unaffected by this issue and can thus outperform GPTAQ under certain INT2 settings.
> >
> > This explains why GPTQ was chosen as the baseline in the INT2 experiments: GPTAQ becomes unstable in the ultra–low-bit regime, and the purpose of our evaluation was to benchmark VLMQ against the strongest reliable baseline under the given quantization setting. **Nonetheless, we emphasize that VLMQ is conceptually orthogonal to the choice of calibration pipeline.** For completeness, we report the results of applying VLMQ to both GPTQ and GPTAQ baselines below.
> >
> > *Average accuracy on Qwen2-VL-7B-Instruct-INT2g128*
> > | Method | Avg ($\uparrow$)|
> > |:------:|:---------------:|
> > | GPTQ | 55.43 |
> > | GPTQ+VLMQ | **57.76** |
> > | GPTAQ | 55.07 |
> > | GPTAQ+VLMQ | **56.51** |
> >
> > *Average accuracy on Qwen2.5-VL-7B-Instruct-INT2g128*
> > | Method | Avg ($\uparrow$)|
> > |:------:|:---------------:|
> > | GPTQ | 55.58 |
> > | GPTQ+VLMQ | **57.46** |
> > | GPTAQ | 41.69 |
> > | GPTAQ+VLMQ | **52.10** |
> >
> > These results clearly demonstrate that VLMQ consistently improves both GPTQ and GPTAQ baselines. Notably, on Qwen2.5-VL-7B-Instruct-INT2g128, VLMQ enhances GPTAQ by approximately 10% in average accuracy. However, because the GPTAQ baseline suffers from extremely poor stability under INT2 settings, the resulting GPTAQ+VLMQ performance is still unable to surpass GPTQ. This further highlights the importance of carefully handling quantization error accumulation in ultra–low-bit scenarios, an aspect we plan to continue investigating in future work.
> >
> > In the revised manuscript, we provide additional discussion to clarify this point.

---

> ### Author Response · Authors · 2025-11-20
>
> > INT3g128: On Qwen2-VL-2B, the average score of VLMQ (62.90) is lower than that of GPTAQ (63.44). The author attributes it to the abnormal performance of GPTAQ on MME RealWorld, but this does not change the fact that VLMQ is not optimal in this setting. INT2g128: On the LLaVA-OneVision-7B model, VLMQ's performance on MMEen (38.66) was significantly lower than GPTQ (40.29) and GPTAQ (42.59), but it surpassed the baseline by a slight advantage on other tasks in the final average score. This situation of "overall victory but lagging behind in key indicators" should be discussed in more depth.
>
> We thank the reviewer for this observation. Regarding the INT3g128 results on Qwen2-VL-2B, the unusually high GPTAQ score on MME-RealWorld appears to arise from method-specific biases and model-dependent factors, although the exact cause is not entirely clear. As this single benchmark even exceeds the full-precision result, it acts as an outlier that disproportionately influences the averaged score. We therefore view this behavior as an isolated instance rather than evidence of a systematic limitation of VLMQ. Such cases of “overall strong performance with occasional dips on individual indicators” are not uncommon in multimodal evaluation, where tasks exhibit different sensitivities to quantization noise. Across the broader range of models and quantization settings presented in the paper, VLMQ consistently provides robust improvements over existing weight-only methods, suggesting that its overall effectiveness is well maintained even if it does not lead on every single benchmark.
>
> ### Reference
> [1] Li, Shiyao, et al. "MBQ: Modality-balanced quantization for large vision-language models." Proceedings of the Computer Vision and Pattern Recognition Conference. 2025.
>
> [2] Lin, Ji, et al. "AWQ: Activation-aware weight quantization for on-device llm compression and acceleration." Proceedings of machine learning and systems 6 (2024): 87-100.
>
> [3] Liao, Baohao, et al. "ApiQ: Finetuning of 2-bit quantized large language model." arXiv preprint arXiv:2402.05147 (2024).
>
> [4] Huang, Wei, et al. "SliM-LLM: Salience-driven mixed-precision quantization for large language models." arXiv preprint arXiv:2405.14917 (2024).

---

### Author Response · Authors · 2025-11-21
**To ACs and Reviewers**

**Dear ACs and Reviewers,**

We sincerely thank you for your time, constructive feedback, and detailed evaluations. We have carefully reviewed all comments and provided comprehensive responses, additional analyses, and new experiments to address each concern. We would like to highlight the following points.

**Strengths Highlighted by Reviewers**

- **Clear motivation and problem formulation.** The paper is well written and clearly identifies two overlooked challenges in VLM post-training quantization: 1)visual over-representation, and 2) the modality gap.
- **Strong methodological clarity, visualization quality.** Visualizations are intuitive and informative, symbol definitions are clear. The block-wise gradient extraction is efficient and well motivated.
- **Comprehensive empirical evaluation.** Extensive experiments and ablation studies validate the effectiveness and robustness of the proposed VLMQ framework.

**Additional Experiments**
- Added an ablation on naive token-weighting with a 3×3 grid (Reviewer RFpm).
- Added compatibility experiments with GPTQ and GPTAQ (Reviewer RFpm).
- Included INT3 results on a larger 32B model to demonstrate scalability (Reviewer x6UG).
- Added broader generalization results on HellaSwag, MMMU, and TextCaps (Reviewers ZiWz and x6UG).
- Added hardware efficiency analysis, including quantization latency and peak memory for 2B/7B/32B models (Reviewers ZiWz and x6UG).
- Included INT4 quantization results (Reviewer x6UG).
- Added analysis on cross-modal alignment and interpretability (Reviewer RFpm).
- Clarified the block-wise backpropagation mechanism and its theoretical foundation (Reviewer RFpm).

**Discussion**
- Explanation for baseline degradation under INT2 quantization (Reviewer RFpm).
- Clarification on the role and limitations of precursor methods (Reviewer RFpm).
- Distinction between our approach and Q-VLM / SliM-LLM (Reviewer ZiWz).
- Analysis of cross-modal alignment and interpretability effects (Reviewer ZiWz).
- Detailed clarification of the block-wise backpropagation mechanism and theoretical rationale (Reviewer ZiWz).
- Discussion on compatibility with QAT, mixed-precision quantization, and streaming scenarios (Reviewer ZiWz).

We respectfully invite the reviewers to consider our revisions and responses. If our clarifications and additional results adequately address the concerns, we would be grateful for your reconsideration. We sincerely appreciate the reviewers’ thoughtful and constructive feedback.

Best Regards,

Authors of Submission #7401

---

### Author Response · Authors · 2025-11-23
**Revised Manuscript Uploaded**

**Dear ACs and Reviewers,**

The updated manuscript has been uploaded. All revisions and newly added materials are clearly marked in **blue** for your convenience. We kindly invite you to refer to the revised version.

Best regards,

Authors of the Submission #7401

---

### Meta-Review · Area_Chair_g7iD · 2026-01-18

**Summary:**

This paper receives 6,4,4 scores, and the main concerns are (1) missing core ablation experiments, (2) unoptimized‌ hyperparameters for AWQ and MBQ, (3) different baselines (GPTQ or GPTAQ) in different experiments, (4) low accuracy on MME-RealWorld, (5) limited Analysis of generalization and robustness (beyond VQA-style), (6) Computational cost, (7) overclaiming integration with other PTQ algorithms. According to the authors' responses, there are many remaining concerns, and I suggest rejection of this paper.

**Reviewer Concerns:**

There are many remaining concerns.
(1) Unoptimized‌ hyperparameters for AWQ and MBQ lead to unfair comparison.
(2) Passing the problems to MME-RealWorld is not convincing
(3) The additional experiment on MMMU is also VQA-style, while it includes only one different style benchmark (TextCaps). The improvement in TextCaps is limited.
(4) Obvious extra computational cost against GPTQ.
(5) Overclaiming integration with other PTQ algorithms.

**Reviewer Scores:**

All reviewers may keep original ratings.

---

### Decision · Program_Chairs · 2026-01-26

Reject